# Acto-myosin force organization modulates centriole separation and PLK4 recruitment to ensure centriole fidelity

Elisa Vitiello [1], Philippe Moreau[1], Vanessa Nunes [2,3], Amel Mettouchi[4], Helder Maiato[2,3,5], Jorge G. Ferreira [2,3,5], Irène Wang[1] & Martial Balland [1]

The presence of aberrant number of centrioles is a recognized cause of aneuploidy and hallmark of cancer. Hence, centriole duplication needs to be tightly regulated. It has been proposed that centriole separation limits centrosome duplication. The mechanism driving centriole separation is poorly understood and little is known on how this is linked to centriole duplication. Here, we propose that actin-generated forces regulate centriole separation. By imposing geometric constraints via micropatterns, we were able to prove that precise acto-myosin force arrangements control direction, distance and time of centriole separation. Accordingly, inhibition of acto-myosin contractility impairs centriole separation. Alongside, we observed that organization of acto-myosin force modulates specifically the length of S-G2 phases of the cell cycle, PLK4 recruitment at the centrosome and centriole fidelity. These discoveries led us to suggest that acto-myosin forces might act in fundamental mechanisms of aneuploidy prevention.

[1] Laboratoire interdisciplinaire de Physique, Université Joseph Fourier (Grenoble 1), Domaine universitaire, Bat. E45 140, Rue de la physique, BP 87, 38402 Saint Martin d'Hères, Cedex 9, France. [2] Chromosome Instability & Dynamics Laboratory, Instituto de Biologia Molecular e Celular, Universidade do Porto, Rua Alfredo Allen 208, 4200-135 Porto, Portugal. [3] Instituto de Investigação e Inovação em Saúde—i3S, Universidade do Porto, Rua Alfredo Allen 208, 4200-135, Porto, Portugal. [4] Institut Pasteur, Département de Microbiologie, Unité des Toxines Bactériennes, Université Paris Descartes, 25-28 Rue du Dr Roux, 75015 Paris, France. [5] Cell Division Group, Experimental Biology Unit, Department of Biomedicine, Faculdade de Medicina, Universidade do Porto, Alameda Prof. Hernâni Monteiro, 4200-319 Porto, Portugal. These authors contributed equally: Irène Wang, Martial Balland. Correspondence and requests for materials should be addressed to E.V. (email: elisa.vitiello@univ-grenoble-alpes.fr)

During cell division, the centrosome has the important role of facilitating mitotic spindle assembly to ensure timely chromosome partitioning between two daughter cells[1–3]. For this reason, a tight regulation between cell cycle and centrosome duplication cycle should be in place[4]. A correct division cycle starts with one centrosome per cell, formed by a pair of centrioles. Centrioles are linked by a proteinaceous bridge mainly composed of c-Nap1 and rootletin[5,6]. During the S phase, new centrioles grow from the parental pair; they elongate and mature in G2, to finally move apart upon cleavage of the parental link[7,8], in order to build the mitotic spindle and guide chromosome segregation.

Even in presence of the proteinaceous link, centrioles have been observed moving apart, although this separation occurs transiently and within a few micrometers[9,10]. The nature and the reason for these movements are still poorly understood. Recent evidences have led researchers to propose that centriole separation might be under the control of cytoskeleton dynamics. This idea was first advanced by Graser et al. showing that the centriolar movement can be regulated by microtubules via the centriolar protein Cep215 and it is interactor pericentrin[11], which serve as anchoring point for microtubules[12,13]. Moreover, centriole separation was recently proposed to impact centrosome duplication rate[14], providing a functional role for this behavior. According to their results, centrioles can initiate duplication at centriole-to-centriole distances up to 80 nm[14]. Higher distances (up to 300 nm) are reached during prophase, suggesting that a duplication block might occur by increasing the distance between the two centrioles[14].

Aberrant centrosome duplication cycles, resulting in more than four centrioles, are one of the main causes of chromosome segregation defects (aneuploidy), a condition highly associated to cancer formation and/or progression[15,16]. Given the strong association between centrosome duplication defects and aneuploidy in several types of cancers[17–21] it is important to understand the mechanisms regulating centrosome duplication.

On the wave of the latest discoveries by Farina et al. showing that purified centrosomes nucleate actin fibers in vitro[22] and by Au et al., reporting new centriolar protein GAS2L1 serving as platform for actin fibers docking[23], we hypothesize that actin-generated forces[24–26] could regulate centriole-to-centriole distance and that this mechanism may be important to ensure correct centriole duplication.

In this manuscript, by imposing geometric constraints via micropatterns we found that acto-myosin forces modulate centriole separation direction, duration and distance. Alongside, we show that inhibition of acto-myosin contractility impairs centriole separation. Moreover, we found that organization of acto-myosin force modulates specifically S-G2 phase length of the cell cycle, PLK4 recruitment at the centrosome and the fidelity of centriole duplication.

## Results

**Acto-myosin forces modulate centriole-to-centriole distance.** Firstly, we monitored the centriole behavior in asynchronous cells. To track centrioles we used HeLa cells stably expressing Centrin1-GFP (C1-GFP). As previously published[9], we observed that in untreated and asynchronous cells, centrioles can transiently separate by a broad range of distances up to 6 µm (Fig. 1a, c). The reason of this transient centriole separation is poorly understood. Taking into account the recent discovery by Farina et al., showing that the centrosome can nucleate actin fibers in vitro, we wondered whether the acto-myosin complex, the main force generator within the cell, could regulate centriole-to-centriole distances. To test this hypothesis, we treated the asynchronous HeLa cell population with 10 µM blebbistatin to inhibit myosin-light chain activity and decrease actin-generated forces. As shown in Fig. 1b, c, contractility inhibition significantly reduces centriole-to-centriole distance. These data suggest that acto-myosin forces contribute to centriole separation.

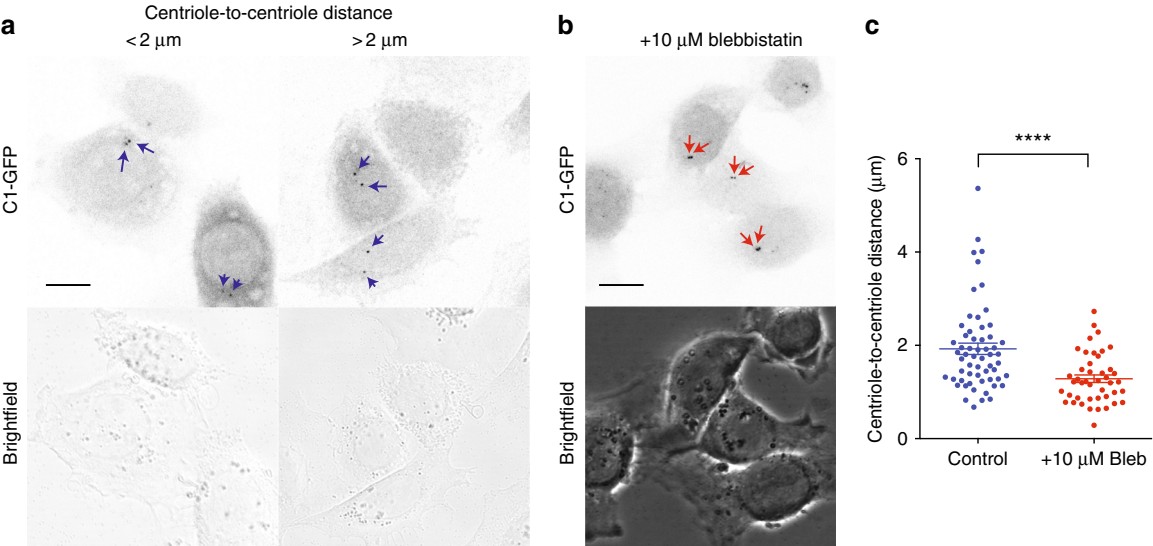

**Fig. 1** Acto-myosin contractility inhibition limits centriole separation in a non-synchronous HeLa cells. **a** Representative pictures of untreated asynchronous HeLa cells expressing Centrin1 (C1-GFP). Blue arrows indicate pair of centrioles. Two different centriole separation behaviors have been observed: in one case (left panel) centrioles separate with distances <2 µm; in the other case (right panel) centrioles separate further than 2 µm, up to 6 µm. Scale bar 10 µm. **b** Inhibition of acto-myosin contractility via blebbistatin treatment significantly reduces the centriole-to-centriole distance. Red arrows indicate pair of centrioles. Scale bar 10 µm. **c** Quantification of centriole-to-centriole distance in non-synchronous HeLa cells, untreated (Control) or treated with 10 µM Blebbistatin (Bleb) (Control $n = 59$ cells; +10 µM Bleb $n = 44$ cells). Error bars represent s.e.m.; $p$-value was obtained with unpaired two-tailed $t$-test; ****$p < 0.0001$

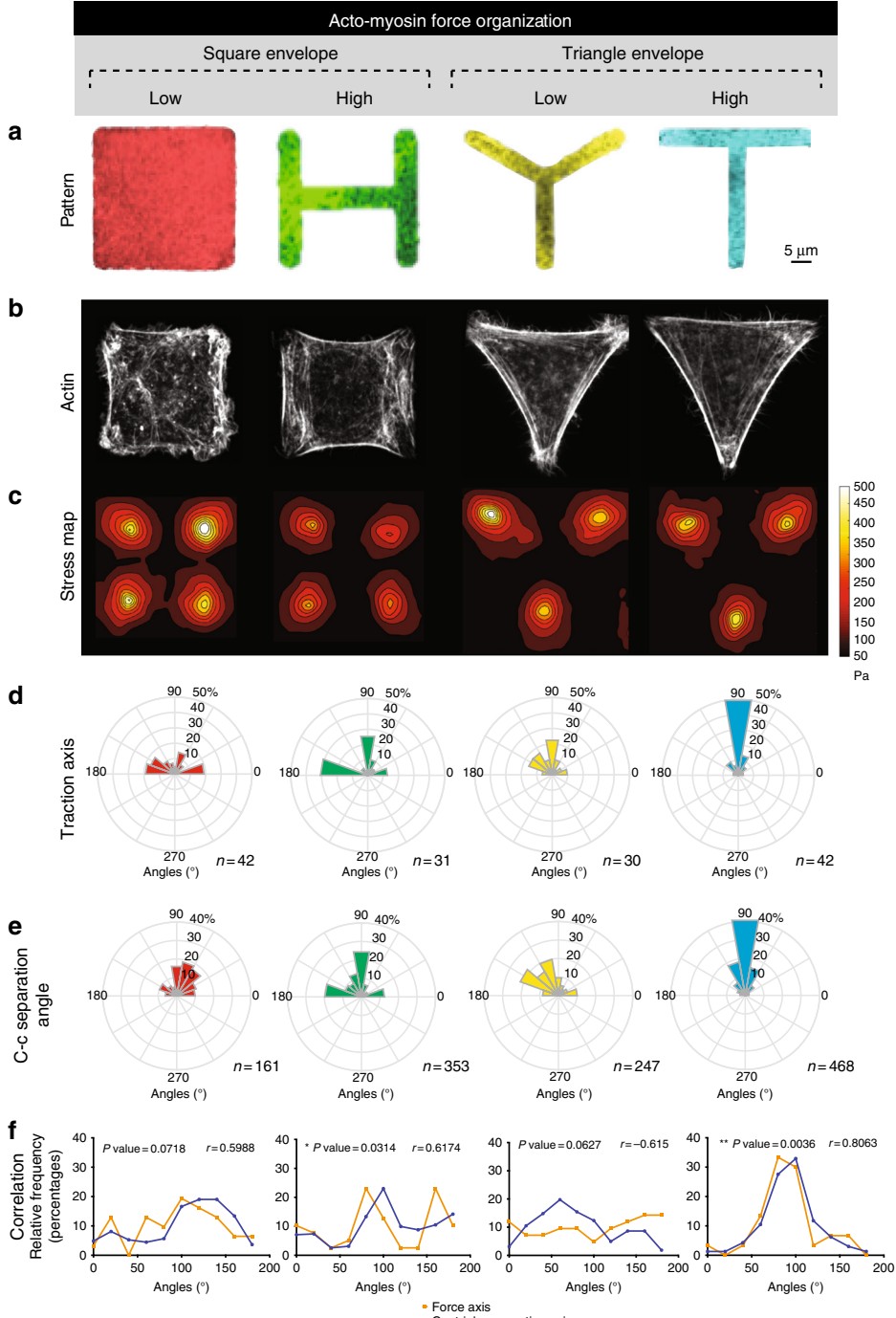

**Fig. 2** Acto-myosin force organization regulates centriole separation. **a** Selected shapes for fibronectin micropatterns printed on polyacrylamide hydrogels: Square in red, H in green, Tripod in yellow, and T in blue. Patterns have equivalent projected area (1000 μm²). **b** Phalloidin staining to decorate actin fibers of Centrin1-GFP (C1-GFP) HeLa cells. **c** Stress map and **d** distribution of principal traction axis calculated for Square (n = 42 cells), H (n = 31 cells), Tripod (n = 30 cells) and T (n = 42 cells). These angles are restricted to the [0°, 180°] range as traction force axes are not directed. **e** Angle histogram of centriole-to-centriole axis (c-c separation angle). These angles are restricted to the [0°, 180°] range as centriole-to-centriole separation axes are not directed. **f** Correlation of angle distribution for Traction axis and c-c separation angle. Pearson test was used to estimate the correlation coefficient (r) and the statistical significance. For the r of H and T, p-values are 0.0314 and 0.0036 (scale bar = 5 μm)

**Organization of acto-myosin forces regulates centriole separation.** Building upon this finding, we decided to test whether modulating the spatial organization of actin fibers and the corresponding traction force orientation could impact centriole separation. Our strategy consisted in using fibronectin adhesive micropatterns, to trigger specific acto-myosin arrangements by imposing a defined geometric constraint[27]. Our micropatterns are

printed on 40KPa hydrogel to mimic physiological microenvironment rigidities[28]. We studied two sets of shapes with equal projected area (Fig. 2a and Supplementary Fig. 1A, B) to decouple the contributions of cell shape and cytoskeleton organization: two squares (Square and H) and two triangles (Tripod and T). We tested the effect of these geometrical constraints on HeLa cells expressing C1-GFP, which allow us to characterize the

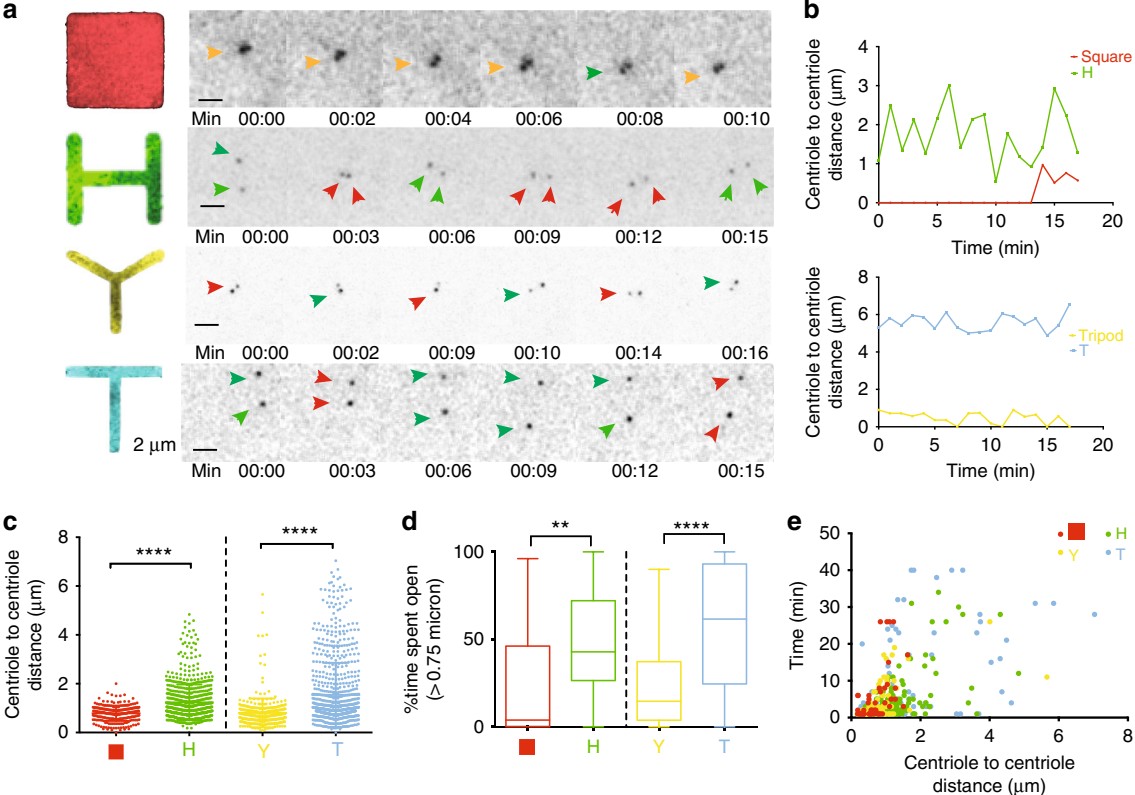

**Fig. 3** Acto-myosin force organization differentially controls dynamics of centriole separation. **a** Selected frames from time-lapse movies of cells expressing C1-GFP (Scale bar = 2 μm). Orange arrows indicate centrioles that maintain a stable distance over time, green indicates an increase in distance, red indicates a decrease in distance. Time is in minutes (min). Images correspond to Supplementary Movies 1–4. **b** Centriole-to-centriole distance plotted over time for the respective videos. **c** Centriole-to-centriole distance for all cells and time points (20–30 time points in each cell). **d** Percentage of time spent open (a centrosome is defined as "open" if centriole-to-centriole distance is above 0.75 μm). **e** Open time versus centriole-to-centriole distance for all the cells imaged on the micropatterns (Square n = 23 cells; H n = 46 cells; Tripod n = 39 cells; T n = 37 cells). Error bars represent s.e.m.; p-values were obtained with unpaired two-tailed t-test; ****p < 0.0001: **p < 0.001

impact of acto-myosin forces on centrioles separation. As our data show, each sub-set included two different adhesive surfaces inducing either randomly (Square and Tripod) or highly organized and reproducible (H and T) actin architectures (Fig. 2b and Supplementary Fig. 1C, D). These actin architectures correspond to precise profiles of traction. As measured by Traction Force Microscopy (TFM), H and T shapes were able to induce reproducible force axes (two main ones for H and one for T) (Fig. 2c, d and Supplementary Fig. 2A, B) and as well as a high degree of polarization on individual cell force patterns (Supplementary Fig. 2C, D). On the contrary, Square and Tripod showed less reproducible force axes and lower polarization degree.

After characterizing the mechanical properties of the chosen micropatterns, we investigated the impact of actin-generated force organization on centrosome positioning and dynamics. Firstly, we observed that in the shapes with a high degree of organization (H-T) the centrosome-nucleus axis (detailed explanation of the method used to measure centrosome positioning can be found in Supplementary Fig. 3A) correlates with the force axes measured by TFM (Supplementary Fig. 4A–C, correlation in Supplementary Fig. 4D). These data suggest that traction forces play a role in centrosome-nucleus axis orientation. Similarly, Théry et al. showed that micropattern geometry impacts centrosome-nucleus axis orientation[29].

Next, we assessed the direction of centriole separation by time-lapse microscopy. Here, we observed that in the case of high mechanical polarization (H and T), the force axis correlated with the axis of centriole separation (Fig. 2e, f). Interestingly, high

correlation of the two axes was found for H and T, whereas for the shapes with low mechanical polarization (Square and Tripod) we could not find a significant correlation (Fig. 2f). These data suggest that acto-myosin forces tend to separate centrioles along their main contraction direction.

We then measured the centriole separation distance range for the cells on the different patterns and we observed that centrioles separate with a distance range below 2 μm on Square and Tripod, and up to 6 μm on H and T (Fig. 3a–c and Supplementary Movies 1–4). Moreover, this separation is sustained for longer times in the case of H and T (Fig. 3d, e and Supplementary Fig. 5A). These results suggest that cells with high degree of acto-myosin force organization separate centrioles for larger distances and longer times.

**Inhibition of acto-myosin forces impairs centriole separation.**
To confirm that actin-generated forces govern centriole dynamics, we inhibited actin contractility by blocking myosin II with blebbistatin (bleb). As shown in Fig. 4, cells plated on H and T, when treated with bleb, were no longer capable of separating centrioles further than 2 μm, similar to what is seen for the low mechanically polarized cells (Square and Tripod) (Fig. 4e–h, Supplementary Fig. 5B and Supplementary Movies 5–6). Similarly, when cells on H and T were treated with ML7 (another commonly used contractility inhibitor) the centriole-to-centriole separation distance was significantly reduced (Fig. 4e–h, Supplementary Fig. 6A–E and Supplementary Movies 7–8). Similar results were obtained in fixed cells (Supplementary Fig. 7a–d).

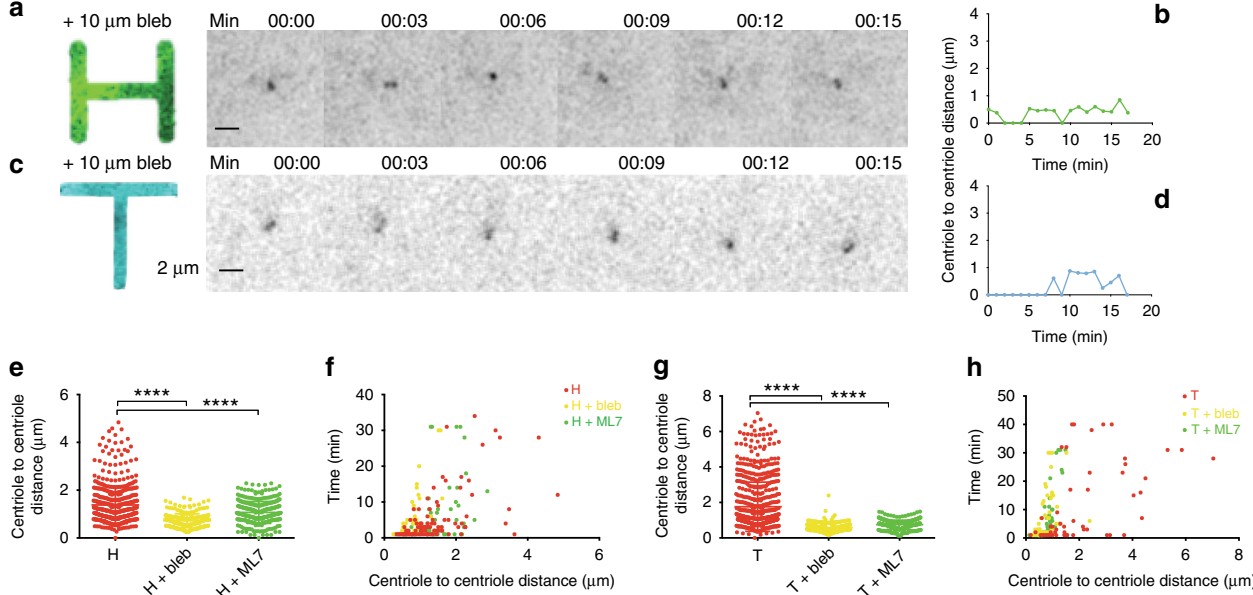

**Fig. 4** Actin contractility inhibition impairs centriole separation. Representative videos of C1-GFP HeLa cells plated on H (**a**) and T (**c**), treated with 10 μM blebbistatin (bleb). Time is in minutes (min). Images correspond to Supplementary Movies 5–6. **b**, **d** Centriole-to-centriole distance plotted over time for the respective videos. Centriole-to-centriole distance for all cells and time points (**e**, **g**) and open time versus centriole-to-centriole distance (**f**, **h**) for control, blebbistatin (bleb) or ML7 treated (both myosin ATP activity inhibitors) cells on H and T patterns (H control n = 46 cells; H + bleb n = 16 cells; H + ML7 n = 18 cells; T control n = 37 cells; T + bleb n = 22 cells; T + ML7 n = 21 cells). Time is in minutes (min). Error bars represent s.e.m. p-value was obtained with one-way Anova test. ****p < 0.0001. Scale bar: 2 μm

Altogether these data show that actin contractility is required for centriole separation in space and time.

The microtubule role in centrosome separation and positioning during mitosis is well characterized[30]. For this reason, we decided to analyze the contribution of microtubule organization and activity in the regulation of centriole separation. To address this question, we plated cells on H and T shape to induce the largest centriole-to-centriole distances. We then interfered with the microtubule network by treating C1-GFP HeLa cells with a low dose of Nocodazole (20 nM), in order to affect microtubule dynamics without completely depolymerizing the network[31]. Upon treatment, we followed the centriole-to-centriole distance over time. As shown, in our setup microtubule dynamics impairment using Nocodazole does not significantly affect the centriole-to-centriole distance (Supplementary Fig. 8A–D, Supplementary Fig. 8F–I, and Supplementary Movies 9–10), or the time the two centrioles stay apart (Supplementary Fig. 8E and J), when compared to control cells (Fig. 3).

**Acto-myosin force organization modulates S-G2 phase duration.** The reason of centriole separation is poorly understood[32,33]. In 2015, Shukla et al. suggested that higher centriole-to-centriole distances, which they observed in prophase (up to 300 nm) could act as a block to reduplication[14]. On the wave of this idea, we hypothesized that the regulation of centriole separation by acto-myosin forces might ultimately affect the cell cycle and centrosome duplication. To test this hypothesis, we synchronized C1-GFP HeLa cells in G1 with a thymidine double block, and plated them on micropatterns. First, we confirmed that in this phase, centriole-to-centriole distance is indeed modulated by spatial force organization. As can be seen in Supplementary Fig. 9A, B, the centriole-to-centriole distance, measured in G1, depends on the degree of mechanical polarization in the same way as asynchronous cells (Fig. 1b). However, the distance range was reduced. Whereas in highly mechanically polarized asynchronous cells we observed distances up to 6 μm, in G1-arrested cells

centriole-to-centriole distances do not exceed the 2 μm. This suggests that centriole distances are cell-cycle regulated. If this is the case, distances above 2 μm could be specific to other phases of the cell cycle, perhaps S/G2 or very early stages of G1, right after cytokinesis. Interestingly, it was recently shown that contractile forces also vary during the cell cycle, increasing during G1 and peaking at S phase[34,35]. This suggests that not only the organization, but also the level of acto-myosin forces might be playing a role in centriole separation.

It is known that centrioles duplicate in S phase and mature during G2[36]. Hence, we wondered if the different organization of acto-myosin forces could directly affect progression through S and G2 phases. To test this, we utilized a FUCCI HeLa cell line to monitor cell cycle phase progression. As shown in Fig. 5a, b, acto-myosin forces organization do not impact G1 phase but they specifically affect the length of S-G2 phase: in particular, shapes with lower mechanical polarization (Square and Tripod) have significantly longer S-G2 phase, compared to their respective, more mechanically polarized shapes (H and T) (Supplementary Movies 11–14). Strikingly, inhibition of acto-myosin contractility via bleb, causes an extension of S-G2 phase for all the shapes, suggesting that acto-myosin forces are required for the correct progression at this stage of the cell cycle (Supplementary Fig. 10A, B, Supplementary Movies 15–19). Taken together, our results led us to speculate that organization of acto-myosin forces might play a role in centriole duplication.

**Acto-myosin force organization regulates PLK4 recruitment to ensure centriole fidelity.** We then investigated the recruitment of PLK4, a known regulator of centriole duplication. PLK4 is a Serine/Threonine kinase that mostly localizes to the centrosome[37] and is known to activate the centrosome duplication machinery[38] and to limit centrosome amplification[37,39–44]. In fact, PLK4 overexpression has been reported to cause centrosome amplification and to be associated with tissue hyperplasia in mice[44]. Moreover, PLK4 inhibition suppresses the proliferation of

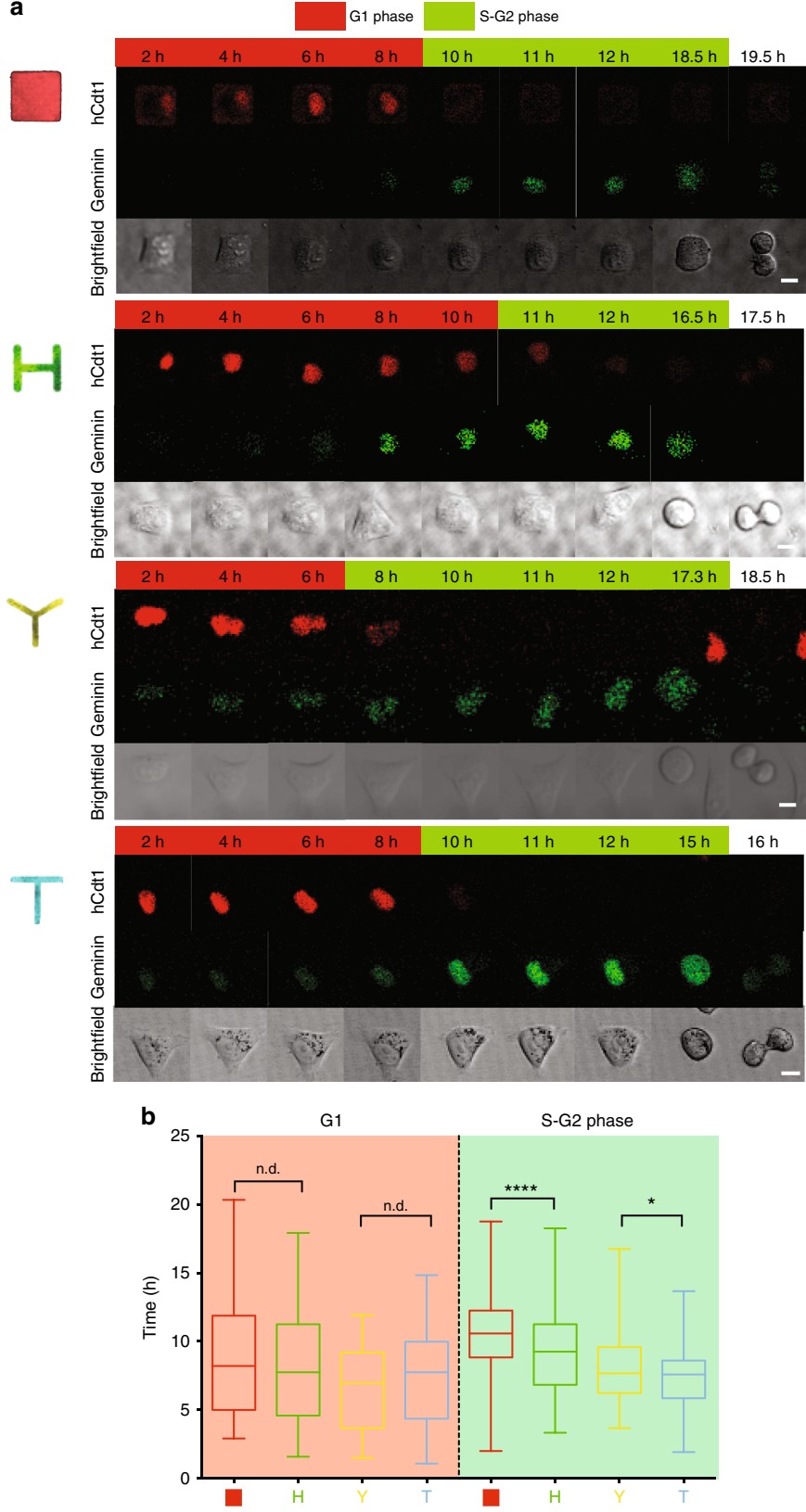

**Fig. 5** Mechanical polarization reduces S-G2 phase duration. **a** Representative hCdt1 (G1-to-S phase reporter in red) and Gemin (S-to-G2 phase marker in green) signal time-lapse for FUCCI HeLa cells plated on respective micropatterns (scale bar = 10 µm). Time is in hours. Images correspond to Supplementary Movies 11–14. **b** G1/S and S-G2 (to NEB: nuclear envelope breakdown) time plot for all the H2B-GFP cells imaged overnight after thymidine release. Whiskers represent minimum and maximum values. Box extends from the 25th to 75th percentiles. Line corresponds to median (Square $n = 90$ cells, Tripod $n = 68$ cells, H $n = 71$, T $n = 53$ cells). Time is in hours. $p$-values were obtained with unpaired two-tailed $t$-test; **$p < 0.01$, ****$p < 0.0001$

patient-derived breast cancer in mice and immortalized cell lines in vitro[45]. This led scientists to propose PLK4 inhibition as possible anti-cancer treatment[46]. Noteworthy, PLK4 has been shown to regulate actin cytoskeleton during cell invasion and metastasis through the Arp2/3 complex[47]. These evidences underline a clear connection between PLK4 and actin, which led us to wonder whether acto-myosin forces might regulate the recruitment of PLK4, and consequently affect centriole duplication.

For this, we stained C1-GFP HeLa cells with pan-antibodies raised against PLK4 (kindly provided by Dr M. Bornens). We observed that less mechanically polarized cells (Square and Tripod) present a significantly higher recruitment of total PLK4 to the centrosome compared to the respective shapes with higher mechanical polarization degree (H and T) (Fig. 6a–c). Overall, these data indicate that acto-myosin forces impact centrosomal recruitment of PLK4, in an inverse manner: the less organized the acto-myosin forces are, the more PLK4 is recruited to the centrosome; the higher the degree of acto-myosin force organization, the less PLK4 is recruited and the higher chances of a bona-fide centriole duplication.

As previously mentioned, high amounts of PLK4 can drive centriole amplification (generation of more than four centrioles). Hence, we wondered whether the shapes recruiting higher levels of PLK4 (Square and Tripod) could generate more centrioles than H and T. We counted Centrin1 dots 10 h after thymidine release (in agreement with the FUCCI data showing this time window as corresponding to duplicating centrioles in S or G2 phase) to assess centriole duplication fidelity. At 10 h from thymidine block release, Tripod, H, and T showed about 70% of cells with duplicated centrioles (four centrioles or more) (Supplementary Fig. 11). On the contrary, cells plated on Square took up to 18 h to reach 70% of cells with duplicated centrioles (Supplementary Fig. 11), suggesting that the low degree of acto-myosin force organization on Square patterns could delay centrosome duplication. In addition, by comparing all the shapes when cells reached 70% of duplicated centrioles, we observed that cells with lower mechanical polarization (and higher PLK4 levels; Square and Tripod) present more incorrect duplication events (more than four centrioles), than the corresponding highly mechanically polarized shapes (with lower PLK4 levels; H and T) (Fig. 6d–f). These results suggest that mechanical polarization affects centriole duplication fidelity by modifying centriolar PLK4 levels. To confirm this finding, we tested whether bleb treatment would impact the effect of acto-myosin contractility on centriole duplication. Due to the prolongation of S-G2 phase observed in the FUCCI HeLa cell line on all the shapes (Fig. 5), we counted the number of duplicated centrioles at 16 h from the beginning of bleb incubation. We observed that contractility inhibition significantly reduces the number of cells with duplicated centrioles on all shapes, as well as the fraction of misduplicated centrioles (more than four centrioles) (Fig. 6d–f). These results suggest that acto-myosin force polarization affects centriole duplication fidelity.

One of the most common ways for cells to accumulate an aberrant number of centrioles is due to cytokinesis failure[48]. To rule out the possibility that the centriole amplification events measured in our experiments were due to cytokinesis failures, we quantified the percentage of cells that failed to complete cytokinesis in H2B-GFP HeLa cells on the different shapes (Supplementary Fig. 12A and Supplementary Movies 19–22). By monitoring H2B-GFP signals and brightfield we could track precisely cell division phases until the formation of the midbody and the repositioning of the two daughter cells on the patterns. In our setup, we found no difference in the frequency of cytokinesis failure among all the shapes (Supplementary Fig. 12A, B).

Cytokinesis failure was observed in <2% of the cases (Supplementary Fig. 12A, B). Considering that Square and Tripod manifest 40% of cells with more than four centrioles, our data suggest that cytokinesis failure cannot be the main cause for the aberrant number of duplicated centrioles in cells with low mechanical polarization.

In addition to centrioles, cells also have centriolar satellites. Centriolar satellites are only detectable in interphase and they dissolve during mitosis[49]. While their role is still not fully understood, they are different entities from duplicated centrioles, which appear during S-phase[49]. Unlike bona fide S-phase duplicated centrioles, centriolar satellites are negative for the Serine/Threonine kinase PLK4[39,50]. Because the extra dots observed in cells with low mechanical polarization could be due to centriolar satellites and not centrioles, we decided to confirm their identity by quantifying the percentage of PLK4-positive foci. We observed an overall low fraction of PLK4 negative foci (20–30%) with no difference between the shapes (Supplementary Fig. 13A–C). This finding indicates that even in the cases where centrioles are overduplicated (as for the Square and Tripod), most of these foci are PLK4-positive, hence not centriolar satellites.

Following up on our previous observations and if centriole duplication is indeed amplified by the higher recruitment of PLK4 in low mechanically polarized cells (Square and Tripod), we wondered if we could reduce the number of aberrant centrioles by inhibiting PLK4 activity. First of all, we checked that inhibition of PLK4 via its specific inhibitor CentrinoneB (CenB)[51] does not affect PLK4 localization at the centrosome per se (Supplementary Fig. 14A–C), indicating that PLK4 localization is mostly independent of PLK4 kinase activity. Next, we quantified the number of centrioles in Square and Tripod following a 16-h incubation with CenB and we observed that PLK4 inhibition significantly reduces the number of total centrioles. Strikingly, both duplicated centrioles, as well as the fraction of misduplicated centrioles (more than four centrioles) are reduced, suggesting that the high levels of PLK4 in Tripod and Square might be associated to the appearance of extra centriolar dots (Fig. 6g, h). Moreover, the fact that PLK4 inhibition reduced the number of cells with aberrant centrioles strengthens the authenticity of the extra centrioles observed in Square and Tripod, as opposed to centriolar satellites. Previously, PLK4 inhibition was shown to cause satellites dispersion[52], but not elimination and in our experiment we observed a reduction of C1-GFP positive dots, indicating that they are authentic duplicated centrioles.

Altogether these results suggest that the spatial organization of acto-myosin forces play a role in controlling PLK4 recruitment and hence centriole duplication fidelity.

## Discussion

Previous work demonstrated that the centrosome nucleates actin cables in vitro via Arp2/3, suggesting a mechanistic link between actin dynamics and centrosome function[53]. Further evidences were provided by Au et al., when they reported the direct link between actin fibers and centriolar components through GAS2L1[23]. Building on these results, we investigated the effect of actin contractility on centriole behavior. Here, we propose a model where acto-myosin forces act as a new regulator of centriole separation and PLK4 recruitment to ensure centriole fidelity (Fig. 7).

In agreement with previous publications, our data show that acto-myosin forces can orient the centrosome-nucleus axis along the axis of traction, highlighting a clear role of traction forces in cell polarity[29]. Moreover, we show that a high degree of mechanical polarization significantly affects direction and dynamics of centriole separation. Interestingly, centriole

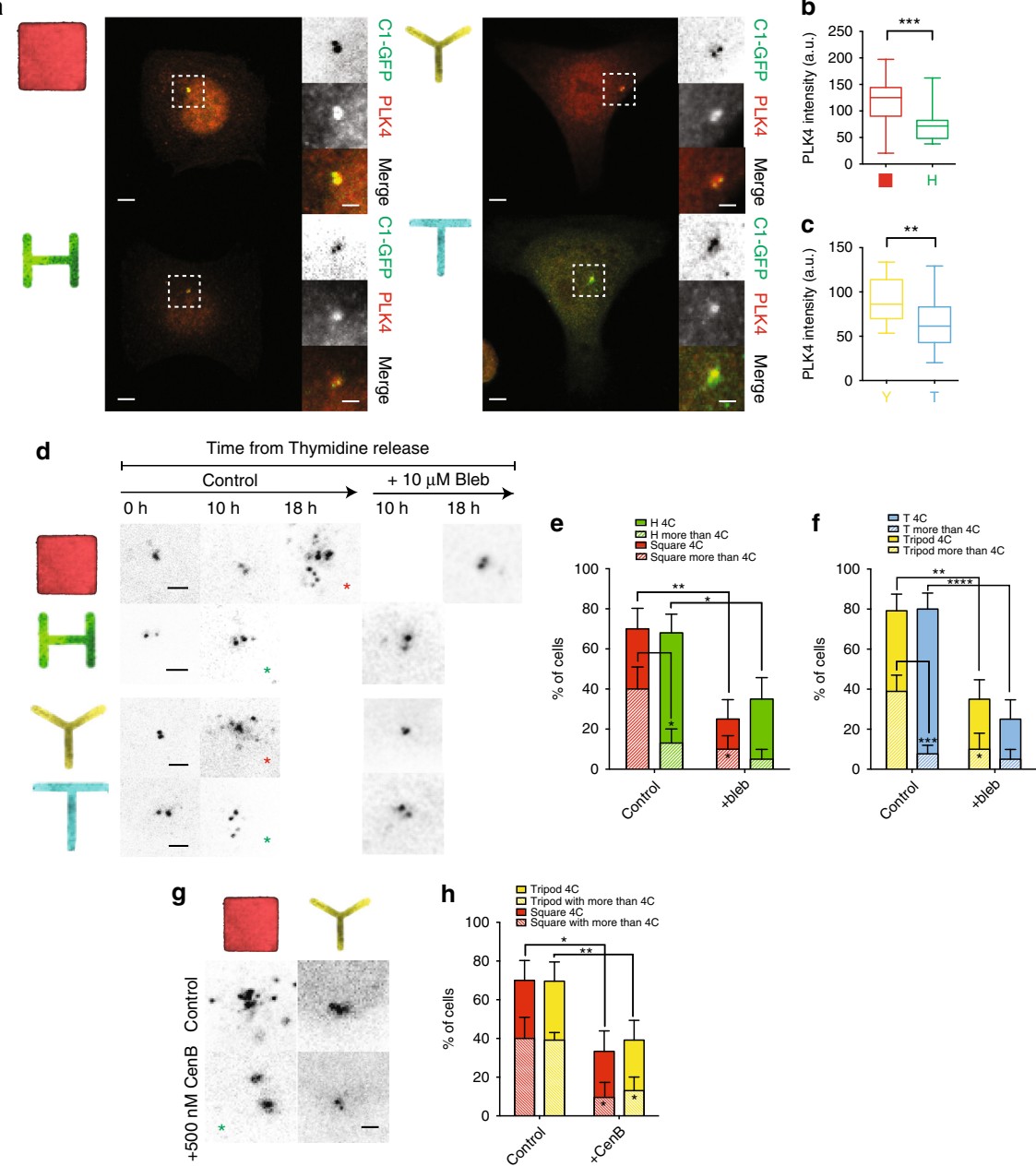

**Fig. 6** Acto-myosin force organization regulates PLK4 recruitment at the centrosome and centriole duplication. **a** Images of cells expressing C1-GFP immunostained for PLK4 (scale bar = 5 μm). Dashed boxes indicate zoomed section presented in panels on the right of the pictures. Zoomed panels (scale bar = 2 μm): from top to bottom: C1-GFP, total PLK4, Merge. **b, c** PLK4 mean fluorescent intensity for all cells analyzed. Between 20 to 30 cells were analyzed for all the conditions. **d** Representative pictures of C1-GFP HeLa cells arrested in G1 phase with a double thymidine block and then analyzed for presence of duplicated centrosomes (4C and more) at different time points from the release of the thymidine block. Cells were either untreated (Control) or treated with 10 μM Blebbistatin (Bleb). Red asterisks indicate misduplicated centrosomes and green asterisks indicate correct duplication. Pictures were analyzed as maximal projection of the z-stack spanning through all the volume of the cell. **e, f** Quantification of duplicated centrioles (4C and more than 4C) at the time point at 10 h from Thymidine release for H, Tripod and T, and 18 h for Square (Square n = 20 cells; H n = 25 cells; Tripod n = 24 cells; T n = 25 cells) and for the corresponding blebbistatin (Bleb) treated cells (Square + Bleb n = 20 cells; H + Bleb n = 20 cells; Tripod + Bleb n = 20 cells; T + Bleb n = 20 cells). Quantification for Square and H shapes (**e**) and Tripod and T (**f**). Error bars represent s.e. **g** Representative pictures of C1-GFP arrested in G1 phase with a double thymidine block and then analyzed after 16 h of CentrinoneB (CenB) treatment versus untreated (scale bar = 2 μm). **h** Quantification of more than four centrioles and more than four centrioles in untreated and 16 h CentrinoneB (CenB) treated Square and Tripod cells (square n = 20 cells; Square + CenB n = 20 cells; Tripod n = 20 cells; Tripod + CenB n = 23 cells). Plain bars represent percentages of cells with four centrioles, patterned bars represent percentage of cells with more than four centriole. * is the statistics (p < 0.05) for treated vs untreated % of cells with more than 4C in Square and in Tripod. Error bars represent s.e. p-values were obtained with unpaired two-tailed t-test; *p < 0.05, ***p < 0.001

separation distance range varies across cell cycle (up to 2 μm in G1-arrested cells; up to 6 μm in asynchronous cells). We did not explore in detail why different phases of the cell cycle have precise centriole separation responses, but this finding raises the

following question: how do acto-myosin forces change during cell cycle? In two very recent publications, the level of traction forces was shown to increase in the G1, reach a plateau in S phase, and then decrease during G2[34,35]. Altogether our data and these

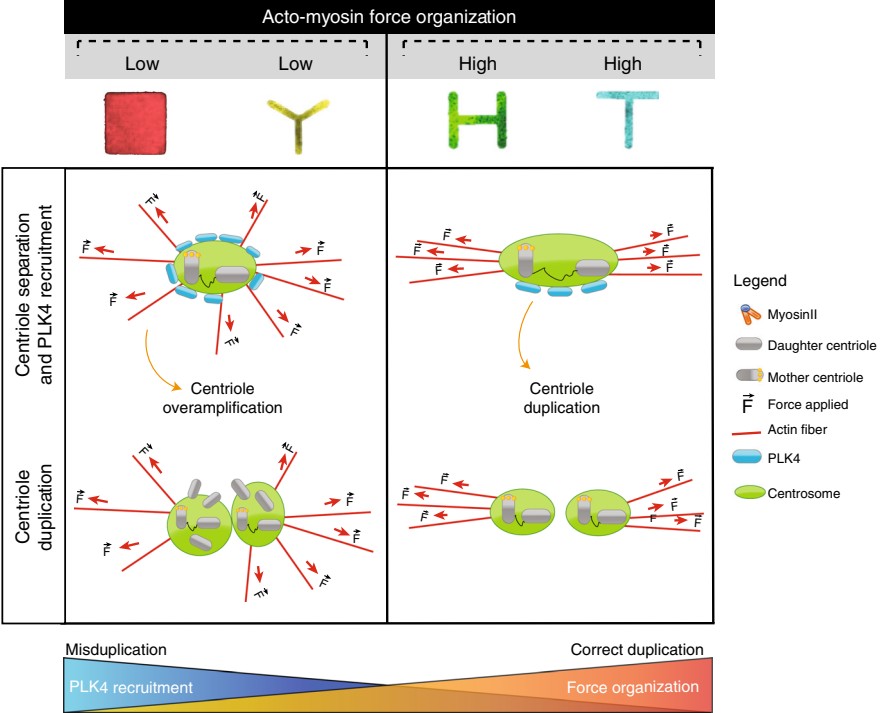

**Fig. 7** Proposed model. We propose here a model in which centrioles sense acto-myosin force organization and respond via regulating centriole separation. Moreover, the degree of acto-myosin organization modulates PLK4 recruitment at the centrosome to ultimately control centriole duplication. In the proposed scenario, the PLK4 recruitment at the centrosome is inversely proportional to the degree of force organization: an excessive recruitment of PLK4 due to disorganization of acto-myosion forces triggers aberrant centriole duplication. On the contrary, well-aligned acto-myosin forces ensures the right amount of PLK4 at the centrosome and the correct centriole duplication

findings suggest that centrioles might be able to respond not only to the organization of acto-myosin network, but also to precise force modulation. To target this question, we speculate that it would be interesting to utilize local force sensors at the centrioles to directly correlate the force sensed with the centriole separation response.

The importance of acto-myosin forces in the cell cycle emerges also from the results of the acto-myosin contractility inhibition experiments. Here, our data show that myosin inhibition leads to a prolonged S-G2 phase and delayed centriole duplication. Previously, Sharma et al. showed that 7–10-day-long treatment of 10 μM bleb causes Wharton's jelly-derived mesenchymal stromal cells (WJ-MSCs) to exit division and arrest in G0[54]. Although this might seem in disagreement with our data, we want to emphasize that the treatment used by Sharma et al. was 7–10 days long, whereas in our case we only treated for 16 h. Interestingly, what caught our attention from their mRNA analysis of bleb treated cells is that, among all the hits, they observed a specific down-regulation of a group of S-phase genes (E2F1, CyclinA, CDC25)[54], the same cell-cycle stage where centriole duplication should occur. Sharma's results in conjunction with ours pave the way to further studies to clarify how acto-myosin forces really affect S-phase gene expression and how this could molecularly impact centriole duplication.

In this article, we propose that PLK4 is capable of sensing the levels of mechanical polarization and responds accordingly to it (Fig. 7). In particular, PLK4 recruitment is inversely proportional to mechanical polarization degree. How PLK4 senses actin organization and contractility level is still an open question. One possible direction to explore would be the role of Arp2/3. The Arp2/3 complex regulates actin polymerization and organizes it into y-branched networks[55]. It has been shown that PLK4 physically binds Arp2/3 and modulates its activity by directly phosphorylating Arp2/

3 at its activation site[47]. Noteworthy, as Farina et al. demonstrated in vitro, Arp2/3 promotes actin nucleation at the centrosome[22]. These evidences made Arp2/3 an attractive candidate to test in the search of the mechanosensing molecule on the centrosome.

PLK4 has been described as key regulator of centrosome duplication by limiting centrosome number[37–42,56]. Here we show that cells with a low degree of acto-myosin force polarization recruit more PLK4 and display an aberrant increase in the number of Centrin1-positive spots when compared to cells with a high degree of mechanical polarization. Since some of these dots may be centriolar satellites, we counted C1-GFP dots that also contains PLK4, which has been shown not to associate with centriole satellites[50] and found that most of them were PLK4-positive (~80%), confirming that they were authentic centrioles. Alongside our results, a further characterization of these dots would be necessary: same dots could be either confirmed by other centriolar marker stainings or via higher resolution imaging techniques such as electron microscopy (EM) to completely confirm the nature of the PLK4-positive dots. Yet, we can here suggest that the extra Centrin1 dots counted in our setup are bona fide duplicated centrioles, since PLK4 inhibition was shown to reduce their number of centrioles. This is in agreement with data previously published: PLK4 inhibition has been reported to cause centriole satellite dispersion but not elimination[52]. Hence, the disappearance of Centrin1-positive dots upon PLK4 inhibition validates the data showing that lower degree of acto-myosin force organization might favor centriole amplification.

Our data potentially link centriole separation with the regulation of centrosome duplication. As observed for cells with more disorganized acto-myosin network (Square and Tripod), centrioles tend to stay close, accumulate an excessive amount of PLK4 and over-duplicate centrioles within the same cell cycle. On the contrary, cells separating centrioles for wider distances and

longer times (H and T) show significantly higher chances to replicate the centrioles only once. In this context, it is tempting to speculate that there might exist a minimal distance that the two new centrioles need to surpass in order to limit centrosome duplication. Similar hypotheses were proposed by Shukla et al., when they showed that mother and daughter centrioles were slightly more separated than the newly generated centrioles, leading them to propose that a critical minimal distance between two centrioles might be responsible of blocking reduplication[14]. Along this line, our results show that centrioles separated for longer time and wider distance have decreased probability to duplicate aberrantly and suggest that above a certain distance centrioles will only replicate once and that they will be protected from further unnecessary duplication cycles. Accordingly, very recent findings by Flanagan et al. show that when centriole splitting is induced by C-Nap1 absence, PLK4 triggered centrosome amplification is prevented[57].

Here we propose a "centriole force sensing" mechanism where acto-myosin forces modulate centriole separation and centrosomal recruitment of PLK4, to ultimately limit centriole duplication at only once per cell cycle. Since the presence of extra centrioles is a recognized hallmark of cancer[20], we suggest that this mechanism acts as a way for the cell to prevent aberrant duplication and limit possible aneuploidy onset. This "centriole force sensing" model opens the way to new strategies of preventing centriole amplification in cancer via targeting actin contractility regulators.

## Methods

**Preparation of micropatterned hydrogels with nanobeads.** To prepare patterned PAA hydrogels, 32 mm coverslips are first plasma cleaned for 30 s and then incubated with a drop of PLL-PEG 0.1 mg/mL in HEPES 10 mM ph 7.4 for 30′ at RT as described in ref. [58].

Afterward, coverslips are put upright to let the excess PLL-Peg run off and placed on a quartz photomask (Toppan) on a 3 µl drop of MilliQ water. The coverslips on the photomask are then exposed to deep-UV for 5′.

After recovery from the photomasks, the coverslips are incubated with 20 µg.ml$^{-1}$ fibronectin (Sigma) and 20 µg.ml$^{-1}$ Alexa546-conjugated fibrinogen (Invitrogen) in PBS for 30′ at RT.

To prepare the gels, a 42 µl drop of 40 KPa mix of Polyacrylamide (Sigma) and bis-acrylamide (ratio described in ref. [59]) is placed onto the fibronectin coated coverslips. A second coverslip of the same size is then placed on top, after previous silanization with a solution of 100% ethanol solution containing 18.5 µl Bind Silane (GE Healthcare Life Science) and 161 µl 10% acetic acid (Sigma) for 5′.

During the polymerization process, the hydrogel adheres to the silanized coverslip and fibronectin proteins are trapped within the acrylamide mash. The silanized coverslip is finally detached by wetting it with MilliQ water and lifting it up with a blade. Hydrogels are stored in PBS at 4 °C.

To perform Traction Force Microscopy, carboxylate-modified polystyrene fluorescent beads (Invitrogen F-8807) are sonicated for 3 min and embedded in the hydrogel during the polymerization process.

**Traction force microscopy imaging and analyses.** For the static TFM experiments, fluorescence beads embedded within the hydrogels are imaged using a 60X oil objective (numerical aperture 1.4) combined with a 1.5 optical multiplier on a Nikon Ti-E microscope with a CCD camera (CoolSNAP HQ2 camera, Photometrics) and controlled with Nikon software. Cells are kept at 37 °C during the imaging.

For live TFM analyses, cells are imaged with a confocal microscope (Leica TCS-SP8) using a 40× objective (oil immersion, numerical aperture 1.3), with a temperature-control chamber set at 37 °C.

Cellular traction forces were calculated using a method previously described[3,4]. Briefly, at each time point, the image of the fluorescent beads embedded in the substrate was compared to a reference image corresponding to a relaxed substrate and taken after washing away the cells. After correcting for experimental drift, the displacement field was obtained by a two-step process consisting of cross-correlation on 9.6 µm sub-images followed by particle tracking to improve the spatial resolution. The final displacement field was interpolated to a regular grid with 1.2 µm spacing. Traction stress reconstruction was performed with the assumption that the substrate is a linear elastic half-space using Fourier transform traction cytometry (FTTC) and zeroth order regularization[5]. The stress map was defined on the same 1.2 µm-period grid. From this stress map and the cell mask, we checked that the out of equilibrium force is <10% of the sum of forces magnitude, as a quality criterion for all cells and time points[6].

The contractile energy, which is the mechanical energy transferred from the cell to the substrate, was computed from the traction map by integrating the scalar product of the displacement and stress vectors over the cell surface. To determine the principal direction of contraction of each cell, we calculated and diagonalized the first moment tensor of the stress[4]. The eigenvector corresponding to the larger eigenvalue gives the direction of the main force dipole. The degree of force polarization is obtained by comparing both eigenvalues. All the calculations are performed in Matlab.

**Cell culture.** HeLa H2B-GFP/α-tubulin-mRFP cell lines were a gift from Patrick Meraldi (University of Geneva, Switzerland). FUCCI HeLa cells were kindly provided by the lab of Yves Usson (University Grenoble Alpes, France).

All the cell lines were cultured at 37 °C and in 5% $CO_2$ atmosphere in DMEM (Life Technologies) medium containing 10% heat-inactivated FBS (Life Technologies) and 100 µg/ml penicillin/Streptomycin (Sigma-Aldrich).

For live imaging, DMEM was replaced by L15 medium (Life Technologies) supplemented with 10% FBS.

Between 100,000 and 50,000 cells were plated on the micropatterned hydrogels. After 1 h, cells were checked for their adhesion to the hydrogels. In case of excessive amount of cells, rinsed with fresh medium to wash off the non-adhered cells. Cells were usually let spread on patterns for 2–4 h.

**Cell synchronization and drug treatment.** Cells were arrested in G1 phase with a double block of for 18 h with 2 mM Thymidine (Sigma-Aldrich, 1:100 from stock 200 mM) as described in ref. [11]. Next, synchronized cells were plated on the micropatterns and fixed for centrosome duplication efficiency analysis when the 70% of cells had duplicated the centrosome[60].

ML7 (Abcam) was used at 10 µM (1:10,000 dilution from 100 mM stock) as indicated in ref. [61]. Blebbistatin was used at 10 µM (Sigma, 1:20,000 from 20 mM stock) to reduce by 80–90% ATPase activity of myosin as shown in refs. [62,63]. CentrinoneB was used at 500 nM (1:4000 dilution from 2 mM stock) as indicated in ref. [51].

**Measurment of cell cycle phase duration.** FUCCI HeLa cells were arrested in G1 phase with a double block of for 18 h with 2 mM. Cells were then plated on patterns and thymidine block release. We measured duration of G1 phase following the signal of hCdt1-RFP (in red in our pictures), marker of G1 phase. The hCdt1-RFP is degraded at the onset of S-phase. Hence, we counted as beginning of S-phase the disappearance of hCdt1-RFP. S-G2 phase is measured as the disappearance of hCdt1-RFP, the increase of the green probe (GFP-hGeminin) till NEB (nuclear envelop breakdown).

**FACS analysis.** The effectiveness of the synchronization using the double thymidine block synchronization in G1/S phase was assessed using FACS. Briefly, DNA content was measured after 4 and 6 h from the release: cells were fixed with cold ethanol 100% for 10′, stained with 40 µg/ml propidium iodide in 0.1% NP-40 and analyzed on a BD FACS Aria flow cytometer.

**Microscopy.** For fixed and live imaging experiments with Centrin1-GFP HeLa cells or H2B-GFP/α-tubulin-mRFP HeLa cells, a Leica TCS SPE confocal microscope with a 40× objective was used. The microscope is controlled through the Leica Application Suite (LAS) X software. Pictures are then processed in Fiji for further quantification in Matlab.

For all the experiments, Hela cells were grown in 6-well plates and on the day of the experiments, between 50,000 and 100,000 cells were plated on 32 mm diameter micropatterned coverslips. After 2 h, cells were checked for spreading. Afterward, cells were either fixed or used for live cell imaging. For fixation, cells were treated for 10 min with 4% PFA diluted in PBS 0.5% Triton X-100. Cells were then washed with PBS (Life Technology) for 10 min and blocked at room temperature for 20 min with a blocking buffer solution containing PBS, 0.5% bovine serum albumin (BSA, Sigma-Aldrich), 0.1% $NaN_3$ (Sigma-Aldrich), and 20 mM Glycine (Sigma-Aldrich).

Actin was stained with 647-fluorescently labeled-phalloidin (1:1000) incubated post-fixation for 1 h in blocking buffer. Fixed cells were then mounted with Mowiol 4-88 (Polysciences, Inc.) onto glass slides and kept at 4 °C overnight before imaging.

Pan-antibodies against PLK4 were a gift by Michel Bornens[41]. All the antibody stainings were performed for 1 h at room temperature, followed by secondary antibody for 1 h.

**Time-lapse microscopy.** For live imaging experiments, micropatterned coverslips are mounted in special chambers built to fit on the Leica SP8 confocal microscope. L-15/10% FBS is used as imaging medium. To image Centrin1-GFP (C1-GFP) expressing cells, a 488 nm laser was used. For the FUCCI cell lines, we use 488 and 561 nm lasers in a sequential scanning mode. For centrosome dynamics and live TFM imaging correlation, 488 and 633 nm lasers were used in simultaneous scanning. All the laser parameters and imaging setups are controlled through the LAS X system.

To image centrosome dynamics, z stacks with 0.7-μm step covering the entire volume of the cell were recorded every 30 s, 1 or 2 min, depending on the type of experiment.

Time-lapse microscopy was performed in an IN Cell Analyzer 2000 (GE Healthcare), using a Nikon 20×/0.45 NA Plan Fluor objective. TL-Brightfield and H2B-GFP channels were acquired every 10 min with temperature set at 37 °C and $CO_2$ at 5%.

**Statistical analysis and data presentation**. The rosette plots were done in Matlab. All the other graphs are plotted with GraphPad Prism (GraphPad Software, San Diego, CA, USA). Standard error of the mean (s.e.m.) was calculated with GraphPad Prism for almost all the graphs. To test if the significance of the results, we use unpaired two-tailed *t*-test for two sample comparison and the one-way analysis of variance (ANOVA) for multiple conditions. To compare each pair of data, we chose Tukey's honestly significant difference test. In the case of proportion graphs, standard error (s.e.) was calculated for each mean, then Chi-square was used to test if two proportion variables were significantly different. For the NC axis and Traction axis angle comparison, we run the Kolmogorov–Smirnov test, specific to comparing two frequency distribution trends.

**Centrosome tracking and centrioles distance measurement**. At each time point, the centrosome was localized in 3D on a z-stack of confocal images. This z-stack is typically 10–15-μm thick with one image every 0.7 μm. The localization is performed in two steps: first, on the maximum Z projection of the image stack, the user clicks on the centrosome (or on both centrioles when they could be separated). Then the location is refined in the XY plane to subpixel accuracy by local maximum detection and centroid calculation. Finally, to localize the centrosome/centriole in Z direction, pixel intensities are integrated, in each image of the stack, over a 250 nm waist Gaussian mask around the previously determined XY position. The resulting profile was fitted by a three-point Gaussian peak to achieve Z localization with a better accuracy than the z-step size. In case the centrioles could not be distinguished in the Z projected image, we checked whether they could be separated in the axial direction by systematically inspecting the profile along Z, obtained around the centrosome as described above. In summary, the centrosome or centrioles were localized with sub-sampling accuracy in 3D. The centrioles separation smaller than the resolution of the microscope (200 nm in XY plane and 800 nm in the axial direction) could not be detected, but for larger separations, the distance between the two centrioles was measured with a typical accuracy better than 100 nm.

The position of the centrosome or of the center point between the two centrioles (when they can be distinguished) was determined relative to the cell nucleus. The nucleus contour was manually drawn. Then both the length and the angle of the vector linking the nucleus centroid to the centrosome were determined. The angle of the oriented nucleus-centrosome vector is relative to the vertical axis of the image, which is also a symmetry axis for all pattern shapes, and spans the whole 360° range.

**Actin orientation analysis**. To calculate the actin order parameter, images of fluorescent actin cytoskeleton were analyzed to determine fibers orientation. First, the user manually draws a region that includes most of the cell inner area while excluding the bright fibers at the cell boundary, which are always oriented according to the pattern. At each pixel of this region, the local orientation was obtained by computing the structure tensor of the image, using a 600 nm-waist Gaussian weighing function centered on the pixel of interest. The pixels of low coherency (the coherency is defined as the ratio of the difference of the structure tensor eigenvalues to their sum) were rejected. Then the average orientation and the order parameter were calculated for each image. The average orientation angle is given by $\bar{\theta} = \arg(\langle \sin\theta \rangle + i\langle \cos\theta \rangle)$ where $\theta$ is the angle between the local orientation and the image vertical axis. The order parameter is $S = \langle \cos(2(\theta - \bar{\theta})) \rangle$.

To calculate the actin orientation, the FibrilTool plugin in ImageJ is used[64].

**PLK4 recruitment quantification**. PLK4 recruitment was quantified from 3D stacks of confocal images where centrosome and PLK4 are simultaneously imaged with two fluorescence channels. Around each centriole position (as determined by the above semi-automatic procedure from the C1-GFP images), mean pixel intensities were computed inside spherical regions of 0.5, 1, 2, 3, 4 μm-radius (beforehand, these regions were convolved with the microscope point spread function which is more elongated in the axial direction). When the centrioles are separated, intensities around both centrioles have been averaged. Average PLK4 intensity at a given distance of the centrosome (e.g., between 0.5 and 1 μm) was obtained by subtracting the signal integrated over the larger sphere (e.g., 1 μm) by the one from the smaller sphere (e.g., 0.5 μm). The radius of 0.5 μm was chosen to show PLK4 recruitment at the centrosome in Fig. 6 and Supplementary Fig. 14.

**Reporting summary**. Further information on experimental design is available in the Nature Research Reporting Summary linked to this article.

## Data availability
Upon reasonable request, data shown in this study are available from the corresponding author.

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

## Acknowledgements

HeLa H2B-GFP/α-tubulin-mRFP cell lines were a gift from Patrick Meraldi. IN Cell time-lapse microscopy was performed at the i3S BioSciences Screening Unit with the assistance of André Maia. Thanks to Michel Bornens (Curie, Paris, France) for PLK4 antibodies, and Monica Bettencourt Diaz for PLK4 and centriole component antibodies. We want to thank key people that made this project possible: Alice Meunier for the important inputs at the real beginning of this project; MOTIV team for the help with any technical issues encountered. Thanks Thomas Boudou and Manuel Théry for all the feedback on this manuscript. This work was funded by grants from the ANR, Arc, PHC-Pessoa Campus France/Fundação para a Ciência e Tecnologia, FEDER - Fundo Europeu de Desenvolvimento Regional funds through the COMPETE 2020 - Operacional Pro-gramme for Competitiveness and Internationalization (POCI), Portugal 2020, and by Portuguese funds through FCT - Fundação para a Ciência e a Tecnologia/Ministério da Ciência, Tecnologia e Ensino Superior in the framework of the project PTDC/BEX-BCM/1758/2014 (POCI-01–0145-FEDER-016589). This work has been partially supported by the LabEX Tec 21 (Investissements d'Avenir: grant agreement No. ANR-11-LABX-0030).

## Author contributions

E.V. analyzed data, wrote this manuscript and performed experiments. V.N. performed experimental work. P.M. setup the microscope for traction force microscopy experiments and provided technical support to this project. A.M. validated at FACS the cell syn-chronization protocol with the double thymidine block. I.W. developed image analysis tools, provided input on the quantitative and statistical analysis of data, wrote part specific sections and supervised this manuscript. J.F. supervised the experiments in Portugal, performed experimental work, supervised the manuscript and suggested important experiments for this paper. H.M. hosted the collaboration in Portugal and provided insightful feedback. M.B. supervised the project and helped with the writing of this manuscript.
