## [Peer Review File · Nature Communications]

Reviewers' comments:

Reviewer #1 (Remarks to the Author):

In this article, Vitiello and colleagues report the very intriguing observation that the distance between centrioles within centrosomes of cultured cells depend on the adhesive pattern cells are plated on. They show that when cells contract, the centrioles split transiently. They then report that cells plated on patterns which induce more centriole separation show less centrin dots (which they interpret as extra-centrioles) and less multipolar division. They propose that centriole separation prevents PLK4 accumulation and thus overduplication. In a last set of experiments, they inhibit the Arp2/3 based actin nucleation, as well as PLK4 activity and show in both case less extra-centrin dots, and less multipolar mitosis, but the interpretation of this experiment is not easy.

As a conclusion, the initial observation reported here, as well as the concept that post-mitotic centriolar splitting - a phenomenon already reported in multiple articles and related to centriole duplication - can be influenced by cell contractility and mechanical polarization and thus by the geometry of adhesion to the extracellular matrix, is both new and really intriguing. This point is well demonstrated in the article and convincing. Other aspects, such as the link with overduplication, mitotic fidelity and cell cycle progression are less convincing. In particular, the causality links are not very clearly established and alternative interpretation of the experiments could be easily proposed. I thus recommend publication after revision of the manuscript.

Main concerns:

1) The article draws a number of conclusions, but both the writing and the presentation of experiments make it sometime difficult to follow the logic. Supplementary figures are particularly confusing, with experiments which seem to be reporting the same thing spread in different figures (for example T and H plus blebistatin, in figures S4, S5 and S12). There is large amount of work to perform on that side.

2) The line of experiments presented by the authors do not always follow a clear logic: the first propose experiments showing that a) specific patterns guide organization of actin fibers and thus contractile polarity of cells – this is fine and already shown by these authors and others; b) on these patterns, centriole splitting is larger for cells with more polarized contractility, and also temporally correlates with phases of larger contractility and depends on Myosin II activity – this is really new and surprising and well presented; c) on patterns with more centriole separation, there are less extra-centrin dots, and less multipolar mitosis – this is interesting, but needs to be strengthened (see below point 3); d) PLK4 staining shows less PLK4 on centrioles in cells on patterns with less splitting; this is also interesting and it follow the line of thought that forces induce splitting which regulates duplication, but also needs to be strengthened (see below point 4); e) the experiments with Arp2/3 and PLK4 inhibitors are then less clear: the authors show that inhibiting PLK4 in turn seem to regulate contractility, meaning the opposite causality link than above...The results shown in figure S10 show a very weak difference, it seems that it is a weak evidence, and not very interesting for the reasoning, or at least confusing. ; f) they then propose that PLK4 and Arp2/3 together couple mechanics and centriole duplication and cell cycle progression, based on very weak evidence. It is not clear that the authors should try to complete the proof of these last points. They should rather concentrate on the initial observations (a to d) and make their proof solid. The conclusion could then be that contractile polarity of cells regulates centriole splitting and thus centriole duplication via the regulation of PLK4 recruitment – this would already be a very exciting and new result, worthy publication in Nature Com.

3) Link between centriole splitting on H and T patterns and centriole duplication: the authors use the centrin-GFP staining to count the number of centrioles. This staining is well known to also label a number of 'centriole satellites' that accumulate in S/G2 and are not additional centrioles. The authors should clarify how they assess the number of centrioles in the different conditions if just based on this staining. If what they look at are satellites and not centrioles, it is still interesting, but then the interpretation of the experiment is different: the adhesive pattern could rather affect the extent of production and/or clustering of these satellites. The authors could make short timelapse recording of these centrioles/satellites to better characterize them on the different

patterns. Another alternative explanation to their observation could then be that the different patterns affect the cell cortical polarity, as previously proposed by M. They and D. Pellman, which would in turn affect the pattern of forces acting on the centrosome (including centrioles and satellites) and thus lead to more or less centrin dots observed in the centrosome region.

4) PLK4 staining on various patterns: this is a very interesting observation, but as presented, it is too preliminary to enable conclusions on the causal link between centriole splitting and PLK4 recruitment. There are two points made: a) there is more PLK4 at centrosomes on patterns which also display less splitting. But there could be other reasons for that than centriole splitting. For example, cells on square or Y patterns could display a different cell cycle progression, when compared to cells on H or T, and thus PLK4 accumulation at centrosomes could be faster in these cells. Could the authors show more directly, on single cells, that more splitting correlates with less PLK4? Figure S10 shows that, on squares, cells with more distant centrioles do have less PLK4, but, for a given distance between centrioles, there is much more PLK4 on squares than on H. How can the authors explain that? b) There is a very striking effect of Arp2/3 inhibition on recruitment of PLK4. In itself, it could probably be the starting point of an entire article, to decipher the mechanisms and the consequences of this Arp/3 dependent recruitment; so having just this result is frustrating and alone, cannot really prove anything useful for the article. Same thing for the rescue of multipolar division. To my knowledge, this observation is new, and potentially very interesting, but impossible to interpret without a large set of additional experiments; is it an effect really at the centrosome, or indirectly at the cell cortex; the worst experiment in that respect is the experiment that shows cell cycle arrest upon blebbistatin treatment; how can we interpret this experiment? Is it the same for cells which are not on patterns or on squares and Y? Is it only for H and T (in which case, what does it mean)? Is it known for non-patterned cells? There are so many possible reasons for that, other than an effect on the centrosome, that it is really not possible to interpret it in the context of this work.

5) Origin of the multipolar spindle/division: it is important that the authors clarify the effect of the patterns they used on centriole/satellites clustering. It is possible that the clustering of centriole/satellites affect the formation of multipolar spindles, as reported by several labs, including the seminal article from Pellman lab, using quite similar adhesive patterns and showing that cells with multiple centrioles tend to form more multipolar spindle in some patterns than others, due to the effect of cortical polarity on centrioles/centrosome clustering. In these articles, the Y pattern produced more multipolar spindles (compared to a bar shaped pattern), like here, but the interpretation was different...The authors could use PLK4 overexpression to induce multiple centrioles and check how they cluster on their various patterns and how they produce or not multipolar spindles. In general, the authors should investigate in more details how the multipolar spindles form in their system. The occurrence of multipolar mitosis is surprisingly elevated on square and Y patterns – much more than what was reported for non-patterned HeLa cells. Why is that the case? The authors could track with more care, on single cells, the formation of these additional spindle poles at mitotic entry, using also the centrin-GFP staining, to check if additional poles appear from splitting of centrosomes or from the additional satellites.

Reviewer #2 (Remarks to the Author):

In this manuscript, Vitiello et al. examine the potential role of actin-based mechanical forces in regulating two aspects of centrosome behaviour—the distancing of the old and new centrioles during interphase, and the ability of the centrioles to duplicate accurately. They conclude that actin-based forces influence both behaviours, and that these forces normally control Plk4 recruitment to the centrosome to prevent centrosome amplification.

There are certainly some very interesting results presented in this study, but I found several aspects confusing and so do not feel that, in its present state, the main conclusions are justified by the data. If these issues can be resolved then I think this study is potentially of sufficient interest to warrant publication in Nature Communications.

Major points:

1. A problem with this study is that it was unclear to me which “state” (high or low mechanical polarisation) the authors regard as “normal” (i.e. how would these cells behave if they were not subject to these artificial shape constraints). Perhaps this is impossible to answer (presumably it depends on the substrate the cells are plated on), but I found it confusing that the mechanically polarised cells exhibited what I thought were unusually large centriole-to-centriole distances, but exhibited no defects in centriole duplication, while the less mechanically polarised cells seemed to exhibit more normal centriole-to-centriole distancing, but then exhibited what appear to be higher levels of centriole duplication errors. This issue should be discussed (it took me a while to realise this, and clearly the increased distancing in interphase has nothing to do with the over-duplication). Ideally, all the centrosome behaviours assayed here should be compared to how the centrosomes behave in these cells under more normal conditions, so we know which behaviours are unusual.

2a. I appreciate the authors efforts to explain their assays, but, as a biologist not used to some of these methods, I still need a bit more help. The stress maps between the Y and T cells shown in Figure 1B look similar, as does the overall organisation of the actin (from the images shown), yet the traction axes are very different. It needs to be explained better why this is so and what this means for the cell.

2b. The authors refer to centrosome positioning, but what they seem to be measuring is the relative positioning of the centrosome and nucleus, and the position of the nucleus seems to vary depending on the traction axis. Thus, the explanation in the text seems an oversimplification of what the authors are really measuring.

2c. It wasn't clear to me how the traction axis actually influenced the “traction” that the centrosome would be “sensing” (which is presumably dependent on both the stress map of the individual cell, and the position of the centrosome within that cell). It seems central to the authors arguments that the centrosomes in these cells are “sensing” different traction environments, but it isn't clear to me that this is true, as perhaps the centrosomes always position themselves within the cell in an area of minimal traction, and this is position depends on the traction axis within the cell, and the traction levels may not be that different at this minimal point. The authors need to more clearly explain how they know that the centrosomes are experiencing different traction environments.

3. The authors conclude that actin contractile forces play a direct role in centrosome separation. To me, this implies that in normal cells, actin contractility is required for centrosomes to separate. I think this is overstating the data. Several studies have indicated that several mechanisms contribute to centrosome separation, and the authors data seems to suggest that in cells without a strong traction axis the effect of perturbing the actin is relatively mild (Figure S5C): it is only in the cells with a high traction axis that actin has a role in driving the greater distancing of the centrosomes that is observed in these cells.

4. The authors conclude that mechanical polarisation regulates centrosome duplication (Figure 4). I think that this is again too strong. I am willing to believe that one can perturb centrosome duplication by forcing cells to adopt fixed geometries, but this is not the same as showing that mechanical polarisation influences centrosome duplication in normal, unperturbed cells. Moreover, some of this data is poorly documented. Figure 4A is headed “time from Thymidine release” and two time points are shown “0h” and “Duplication”. On reading the legend it seems that this second “Duplication” timepoint is the time at which “all the shapes reach 70% of centrosome duplication”. I am not sure exactly what this means or how this was scored (especially as the authors are claiming that many cells have extra centrosomes). Moreover, it seems possible to me that the cells might have different cell cycle dynamics, so is each shape scored independently (in which case this

second time point might be a different actual time point for each cell shape) or is the 70% figure an amalgamation of all the cells (in which case the different cell shapes might be at different time points in their cell cycles).

5. Another potential problem with this centrosome duplication analysis is that it is well documented that centrin-GFP can form transient assemblies, that are not centrioles, in cells under certain conditions. Perhaps mechanical polarisation can influence the tendency of centrin-GFP to form these transient dots? I think the authors need to prove more thoroughly that these centrin-dots are really bona-fide centrioles (ideally with EM, but at the very least with other markers, and by showing that these extra centrosomes are stably inherited by the daughter cells after cell division).

Minor Points:

1. I was confused about the data showing the quantification of centrosomal Plk4 levels. In Figure 5A the images all show relatively high levels of centrosomal Plk4, and quantify the strong decrease in Plk4 levels observed after various drug treatments. In Figure S9 (mistakenly referred to as 5E in the text?), there is very little Plk4 detected in any of the images shown, and the authors show that the amount of centrosomal Plk4 actually depends on the position of the centrosome within the cell. Why the difference in these images, and how do the authors account for this positional difference when they quantify the amount of Plk4 at the centrosomes in Figure 5A?

2. The signal to noise seems very low in the Y cells shown in Figure 2A. This should be commented on.

In summary, even if the authors can address these issues, I would advise a more caution in their headline claims. These findings might suggest that cells in which mechanical polarisation has been imposed in this relatively artificial system have a tendency to separate to a greater degree and perhaps to over-duplicate, but this does not mean mechanical polarisation normally plays a major part in regulating centriole separation and duplication in normal tissues. Thus, the results reported here are an interesting first step in understanding the potential link between mechanical polarisation and centrosome behaviour, but they do not prove this link normally plays a major role in regulating centrosome behaviour in normal cells.

Reviewer #3 (Remarks to the Author):

This paper is very poorly written and difficult to follow because of its careless use of language in the main body of the text. Throughout the paper, we are told that "cells" respond to various treatments and the reader is only told of the identity of these cells a single time in the Results section. This is not a minor point, as the study focuses on one tumour cell type and so its generality is uncertain. Moreover, phenomenon are described that we told are affecting centrosomes or centrioles interchangeably ...which? And the phenomenon affected is either centrosome / centriole duplication or sometimes amplification. I am concerned that this lack of precision extends into the interpretation of the data which is largely correlative in its nature.

The conclusion of the first section of the paper is that "cells with high mechanical polarization can separate centrioles for larger distances and longer times". While this is certainly one correlation, there are many others that can be made here because the organisation of the HeLa cells is greatly disrupted by their culture on these micropatterns.

This question of interpretation extends to the second section that uses Blebbistatin to block myosin II. It reaches a conclusion that actin contractility is required for centriole separation. Here perhaps the authors mean centrosome separation as centriole separation is the disengagement process that occurs in early G1. Notwithstanding the question of what they mean, the observation is again only correlative and does not indicate whether the effect observed is direct or indirect. This same

question of interpretation arises with the analysis of the relationship between inter-centriolar (do the authors mean centrosomal?) distance and cellular traction.

The authors also attempt to test the relationship between actin generated forces and “centriole duplication number”.... Although I don't understand what they mean here. Here they show, not unexpectedly, that centriole/centrosome separation depends upon the cell cycle phase placing some degree of doubt upon their earlier experiments with unsynchronised cells. Their conclusions about the fidelity of centriole duplication go beyond the reasonable interpretation of their data as they do not examine the centriole duplication cycle in any way whatsoever.

They then attempt to draw conclusions based upon the enrichment of Plk4 at the centrosomes (Do they mean centrioles?) without any regard of the existing knowledge of Plk4 recruitment in the centriole duplication cycle. The authors should relate their findings to the known ability of Plk4 to re-localize from a ring around the outer part of the centriole to a single dot like structure on procentriole formation – the key step in the centriole duplication cycle and whether the effects they observe are similar, for example, to those seen when Pll4 levels increase as a consequence of inhibition of the SCF complex.

The final section about potential roles of Arp2/3 in Plk4 recruitment is simply unconvincing.

In conclusion, the authors should re-evaluate their manuscript both with respect of the precision with which they refer to the structures being studied and the level of analysis of functions for which there is already a very solid basal level of knowledge to which any new findings should be referred.

RESPONSES TO REVIEWER'S COMMENTS

Reviewer #1 (Remarks to the Author):

In this article, Vitiello and colleagues report the very intriguing observation that the distance between centrioles within centrosomes of cultured cells depend on the adhesive pattern cells are plated on. They show that when cells contract, the centrioles split transiently. They then report that cells plated on patterns which induce more centriole separation show less centrin dots (which they interpret as extra-centrioles) and less multipolar division. They propose that centriole separation prevents PLK4 accumulation and thus overduplication. In a last set of experiments, they inhibit the Arp2/3 based actin nucleation, as well as PLK4 activity and show in both case less extra-centrin dots, and less multipolar mitosis, but the interpretation of this experiment is not easy. As a conclusion, the initial observation reported here, as well as the concept that post-mitotic centriolar splitting - a phenomenon already reported in multiple articles and related to centriole duplication - can be influenced by cell contractility and mechanical polarization and thus by the geometry of adhesion to the extracellular matrix, is both new and really intriguing. This point is well demonstrated in the article and convincing. Other aspects, such as the link with overduplication, mitotic fidelity and cell cycle progression are less convincing. In particular, the causality links are not very clearly established and alternative interpretation of the experiments could be easily proposed.

Reply: We agree with the reviewer on this point. Therefore, we decided to take out the part dedicated to cell cycle progression and aneuploidy. This, per se, would require further detailed investigations, that would represent a complete separate project. Instead, we decided to focus on the main message of the paper: how acto-myosin forces impact centriole separation and how this impacts centriole duplication via PLK4 recruitment at the centrosome.

I thus recommend publication after revision of the manuscript.

Main concerns:

1) The article draws a number of conclusions, but both the writing and the presentation of experiments make it sometime difficult to follow the logic. Supplementary figures are particularly confusing, with experiments which seem to be reporting the same thing spread in different figures (for example T and H plus blebbistatin, in figures S4, S5 and S12). There is large amount of work to perform on that side.

Reply: We have worked on the text to make it clearer and more fluid. We have changed the figure order. We believe that that the confusion caused by the previous text is now solved. For instance, S4 is now a main figure (that represents live imaging experiments for centriole-centriole separation (now figure4)). S5 represented the measurements of centriole separation in fixed samples. We left

this one in the supplementary data. S12 showed the centriole separation upon blebbistatin treatment after 16hours. We agreed this is redundant and it has been taken out of the paper.

2) The line of experiments presented by the authors do not always follow a clear logic: the first propose experiments showing that a) specific patterns guide organization of actin fibers and thus contractile polarity of cells – this is fine and already shown by these authors and others; b) on these patterns, centriole splitting is larger for cells with more polarized contractility, and also temporally correlates with phases of larger contractility and depends on Myosin II activity – this is really new and surprising and well presented; c) on patterns with more centriole separation, there are less extra-centrin dots, and less multipolar mitosis – this is interesting, but needs to be strengthened (see below point 3); d) PLK4 staining shows less PLK4 on centrioles in cells on patterns with less splitting; this is also interesting and it follow the line of thought that forces induce splitting which regulates duplication, but also needs to be strengthened (see below point 4); e) the experiments with Arp2/3 and PLK4 inhibitors are then less clear: the authors show that inhibiting PLK4 in turn seem to regulate contractility, meaning the opposite causality link than above...The results shown in figure S10 show a very weak difference, it seems that it is a weak evidence, and not very interesting for the reasoning, or at least confusing. ; f) they then propose that PLK4 and Arp2/3 together couple mechanics and centriole duplication and cell cycle progression, based on very weak evidence. It is not clear that the authors should try to complete the proof of these last points. They should rather concentrate on the initial observations (a to d) and make their proof solid. The conclusion could then be that contractile polarity of cells regulates centriole splitting and thus centriole duplication via the regulation of PLK4 recruitment – this would already be a very exciting and new result, worthy publication in Nature Com.

Reply: We appreciate the reviewer for the enthusiasm shown on our results and we agree again with the reviewer on this point and this is why we present here a version of the paper that focused on centriole dynamics and PLK4 with no reference to Arp2/3. Data on Arp2/3 and its involvement in PLK4 mediated centriole duplication in response of acto-myosin forces is still preliminary and it could be the subject of a dedicated project. We have though mentioned in the discussion that there are evidences in literature suggesting that it would be interesting focusing on Arp2/3 role in PLK4 mediated centriole duplication and that this could be a perspective of continuation of this project. We have now added a full characterization on all the shapes of how inhibition of contractility via blebbistatin inhibits centriole duplication, and prevent cell cycle progression. Moreover we show that PLK4 kinase activity is not necessary to recruit PLK4 at the centrosome (New Fig 6B,C and Fig. S14), and that aberrant centriole duplication in cell with low degree of acto-myosin organization can be rescued by inhibition of PLK4 (Fig. 7).

3) Link between centriole splitting on H and T patterns and centriole duplication: the authors use the centrin-GFP staining to count the number of centrioles. This staining is well known to also label a number of 'centriole satellites' that accumulate in S/G2 and are not additional centrioles. The authors should clarify how they assess the number of centrioles in the different conditions if just based on this staining. If what they look at are satellites and not centrioles, it is still interesting, but then the interpretation of the experiment is different: the adhesive pattern could rather affect the extent of production and/or clustering of these satellites. The authors could make short timelapse recording of these centrioles/satellites to better characterize them on the different patterns. Another alternative explanation to their observation could then be that the different patterns affect the cell cortical polarity, as previously proposed by M. They and D. Pellman, which would in turn affect the pattern of forces acting on the centrosome (including centrioles and satellites) and thus lead to more or less centrin dots observed in the centrosome region.

Reply: We thank the reviewer for raising this issue. To elucidate the nature of the centrin1 dots, we have characterized their ability to recruit PLK4: we have added a completely new figure (FigS13A-C), showing that about 80 per cent of the dots counted are decorated by PLK4. As stated in "Centriolar satellite- and hMsd1/SSX2IP-dependent microtubule anchoring is critical for centriole assembly" by Hori et al (Mol Biol Cell, 2015), centriole satellites do not recruit PLK4, indicating that PLK4 can be used as a centriole marker to distinguish them from satellites. From our experiments we observed that some of the counted spots were centriole satellites (in 20% of the cells), but that this percentage was not different among the shapes analysed, suggesting that the differences in centriole number observed among shapes with high vs low acto-myosin order is not due to the presence of more centriole satellites, but rather to over duplication of centrioles.

4) PLK4 staining on various patterns: this is a very interesting observation, but as presented, it is too preliminary to enable conclusions on the causal link between centriole splitting and PLK4 recruitment. There are two points made: a) there is more PLK4 at centrosomes on patterns which also display less splitting. But there could be other reasons for that than centriole splitting. For example, cells on square or Y patterns could display a different cell cycle progression, when compared to cells on H or T, and thus PLK4 accumulation at centrosomes could be faster in these cells. Could the authors show more directly, on single cells, that more splitting correlates with less PLK4? Figure S10 shows that, on squares, cells with more distant centrioles do have less PLK4, but, for a given distance between centrioles, there is much more PLK4 on squares than on H. How can the authors explain that?

Reply: We did not observe a direct correlation between centriole separation and PLK4 recruitment on single cells. We believe that the origin of the confusion lays in the previous Fig. S10: This plot did not show the intensity versus the distance between centrioles, but it aimed to show the level of PLK4 at difference distances from the centriole signal centre, to indicate that the PLK4 value we measured is really at the centriole area. This figure is no longer in the present version since it was confusing and not very informative.

We believe that PLK4 recruitment is not driven by centriole separation, but rather that both processes are controlled by acto-myosin force organization. Fig 6 shows that more ordered actin organization induces lower level of PLK4 recruitment.

b) There is a very striking effect of Arp2/3 inhibition on recruitment of PLK4. In itself, it could probably be the starting point of an entire article, to decipher the mechanisms and the consequences of this Arp/3 dependent recruitment; so having just this result is frustrating and alone, cannot really prove anything useful for the article. Same thing for the rescue of multipolar division. To my knowledge, this observation is new, and potentially very interesting, but impossible to interpret without a large set of additional experiments; is it an effect really at the centrosome, or indirectly at the cell cortex; the worst experiment in that respect is the experiment that shows cell cycle arrest upon blebbistatin treatment; how can we interpret this experiment? Is it the same for cells which are not on patterns or on squares and Y? Is it only for H and T (in which case, what does it mean)? Is it known for non-patterned cells? There are so many possible reasons for that, other than an effect on the centrosome, that it is really not possible to interpret it in context of this work.

Reply: We took out the data regarding Arp2/3 that would require a further and more detailed characterization, but we addressed the point raised by the reviewer in figure 6 and 7. In these more complete figures, we show that the recruitment of PLK4 is due to acto-myosin force organization (Figure 6) and that this pool is independent of PLK4 activity itself. PLK4 senses the acto-myosin force organization and is more or less recruited at the centrosome, accordingly. Inhibition of PLK4 activity does not impair PLK4 localization (Fig 6 and Fig S14).

To address the reviewer's question about the meaning of cell cycle arrest upon blebbistatin treatment, we undertook further experiments and confirmed the role of acto-myosin forces in centriole duplication: we looked at cells on Square and Tripod to for a better understanding of the process, and as observed already for H and T we quantified a significant reduction of duplicated centrioles in Square and Tripod (added Fig S11A). These data suggest that contractility is necessary for cell cycle progression. Our cells were synchronized in G1 and treated with blebbistatin at the moment of the release. We now included in the paper a comment about how our data are in agreement with evidences presented by Sharma et al, who showed that 10 μ M blebbistatin treatment -same concentration we used for our experiments- leads to G0/G1 arrest in Wharton's jelly-derived mesenchymal stromal cells (WJ-MSCs) from umbilical cord (Sharma et al, Cytotherapy, 2014). According to our data (added Fig6) blebbistatin treatment reduces the number of amplified centrioles in Square and Tripod, indicating a specific role for acto-myosin contractility in centriole duplication.

We did not investigate in details if PLK4 responds directly to the force or indirectly. We believe that this would require as suggested by the reviewer a long list that would promote an independent research project.

5) Origin of the multipolar spindle/division: it is important that the authors clarify the effect of the patterns they used on centriole/satellites clustering. It is possible that the clustering of centriole/satellites affect the formation of multipolar spindles, as reported by several labs, including the seminal article from Pellman lab, using quite similar adhesive patterns and showing that cells with multiple centrioles tend to form more multipolar spindle in some patterns than others, due to the effect of cortical polarity on centrioles/centrosome clustering. In these articles, the Y pattern produced more multipolar spindles (compared to a bar shaped pattern), like here, but the

interpretation was different...The authors could use PLK4 overexpression to induce multiple centrioles and check how they cluster on their various patterns and how they produce or not multipolar spindles. In general, the authors should investigate in more details how the multipolar spindles form in their system. The occurrence of multipolar mitosis is surprisingly elevated on square and Y patterns – much more than what was reported for non-patterned HeLa cells. Why is that the case? The authors could track with more care, on single cells, the formation of these additional spindle poles at mitotic entry, using also the centrin-GFP staining, to check if additional poles appear from splitting of centrosomes or from the additional satellites.

Reply: We took out the data about chromosome segregation, because the best way to address this point would be to perform live imaging on micropatterns to investigate the behaviour of the multiple centrioles during mitosis. We actually tried through the years, to record time-lapses of centrioles in cells plated on patterns, but the task has been always challenging. The difficulty of this experiment relies in the fact that our patterns, as explained in Material and Methods in more details, are printed on polyacrylamide substrates to keep the rigidity of 40Kpa constant. The thickness of the gels plus the micropatterns and the volume of the cell on top represent the limiting factor for the correct and long time-lapse experiments we longed to do. Altogether cells are located on a too high focal plane that can be difficultly reached by the objective working distances. Even when reached, the focus is hardly stable for long time, making impossible recording from single centrioles to duplicated centrioles and mitosis.

In our opinion, the experiment suggested are out of the scope of the new version of the manuscript which is centered on the role of acto-myosin force organization on centriole splitting and duplication.

Reviewer #2 (Remarks to the Author):

In this manuscript, Vitiello et al. examine the potential role of actin-based mechanical forces in regulating two aspects of centrosome behaviour—the distancing of the old and new centrioles during interphase, and the ability of the centrioles to duplicate accurately. They conclude that actin-based forces influence both behaviours, and that these forces normally control Plk4 recruitment to the centrosome to prevent centrosome amplification.

There are certainly some very interesting results presented in this study, but I found several aspects confusing and so do not feel that, in its present state, the main conclusions are justified by the data. If these issues can be resolved then I think this study is potentially of sufficient interest to warrant publication in Nature Communications.

Major points:

1. A problem with this study is that it was unclear to me which “state” (high or low mechanical polarisation) the authors regard as “normal” (i.e. how would these cells behave if they were not subject to these artificial shape constraints). Perhaps this is impossible to answer (presumably it depends on the substrate the cells are plated on), but I found it confusing that the mechanically polarised cells exhibited what I thought were unusually large centriole-to-centriole distances, but exhibited no defects in centriole duplication, while the less mechanically polarised cells seemed to exhibit more normal centriole-to-centriole distancing, but then exhibited what appear to be higher levels of centriole duplication errors. This issue should be discussed (it took me a while to realise this, and clearly the increased distancing in interphase has nothing to do with the over-duplication). Ideally, all the centrosome behaviours assayed here should be compared to how the centrosomes behave in these cells under more normal conditions, so we know which behaviours are unusual.

Reply: To address this concern, we performed new experiments on continuous substrate (without patterns), which are presented in Figure 1. We believe these data show that the situation is not different in not-on-micropatterns cells: we observe that HeLa cells can separate centrioles with the same distance range as shown by the cells on adhesive patterns. We agree that it would be very hard to claim whether low mechanical polarization is abnormal whereas high mechanical polarization is not. Tissue and cells within feel tension via geometric constraint and changes in the morphology or the neighbouring cells. These allow them to polarize accordingly. Disorganization of a tissue causes loss of cell polarity as proven by many groups. We believe that a mechanical polarization has, as we show in the paper, a key role in cell division and in particular in centriole separation and duplication. What we discussed in the end of the paper largely is that mechanical polarization acts to separate the two centrioles, and that this regulates recruitment of PLK4 and limit centriole amplification. The centriole distances observed on micropatterns were confirmed on

cells not on geometrical confinement, indicating that even in more “normal” conditions, centrioles are capable of separating for large distances.

By presenting the data on non-patterned cells at the opening of the new manuscript, we believe that the use of micropatterns is better justified as it enables to investigate the relationship between acto-myosin forces and centriole separation. This also proves that the results observed are not only induced by the artificial constraints imposed to the cells.

2a. I appreciate the authors efforts to explain their assays, but, as a biologist not used to some of these methods, I still need a bit more help. The stress maps between the Y and T cells shown in Figure 1B look similar, as does the overall organisation of the actin (from the images shown), yet the traction axes are very different. It needs to be explained better why this is so and what this means for the cell.

Reply: We thank the reviewer for this question. Traction force microscopy measures the stress (force per unit area) that cells exert on the substrate, which includes both the magnitude and the orientation of these stresses. The stress maps the reviewer refers to show only the magnitude of stresses which spatial distribution, provided the cell envelope is conserved (square or triangle), is mostly unchanged. This supports our message which is that the important factor that drives centriole splitting is not only the magnitude, but rather the orientation of stresses, which is related to the degree of order in acto-myosin organization. To analyse the stress maps in both magnitude and orientation, we calculated the force dipole (or first moment tensor) which enables to find 2 parameters: i) the direction of the main contraction (corresponding to the direction of cell mechanical polarization) shown in new Fig.2D and ii) the difference in contraction level between the main and secondary directions shown in Fig. S2D, which indicates the degree of mechanical polarization within each cell. We found that the shapes H and T induce both more reproducible directions from one cell to another (Fig.2D) and higher degree of mechanical polarization within each cell (Fig. S2D), compared to Square and Tripod.

2b. The authors refer to centrosome positioning, but what they seem to be measuring is the relative positioning of the centrosome and nucleus, and the position of the nucleus seems to vary depending on the traction axis. Thus, the explanation in the text seems an oversimplification of what the authors are really measuring.

Reply: We agree we overstated the message in this section. We now changed the text by specifying we are measuring the centrosome-nucleus axis, and not centrosome positioning. We apologies for the overstatement (fig S4).

2c. It wasn't clear to me how the traction axis actually influenced the “traction” that the centrosome would be “sensing” (which is presumably dependent on both the stress map of the individual cell, and the position of the centrosome within that cell). It seems central to the authors arguments that the centrosomes in these cells are “sensing” different traction environments, but it isn't clear to me that this is true, as perhaps the centrosomes always position themselves within the cell in an area of minimal traction, and this is position depends on the traction axis within the cell, and the traction levels may not be that different at this minimal point. The authors need to more clearly explain how they know that the centrosomes are experiencing different traction environments.

Reply: We agree with the reviewer that we cannot presume to access the forces directly sensed by the centrosome, as this would require local force sensors (e.g. FRET sensors) that do not exist at present. This would be interesting to develop in the future. Here we only measure traction forces the cell exert on the substrate and not forces sensed by the centrosome.

Let us clarify our strategy: we used micropatterning as a tool to modulate the cytoskeletal organization of the cells (this was proved possible in a previous work, Mandal et al., Nat Com, 2014) and study the consequences of this acto-myosin organization on centriole splitting and duplication. Measurement of traction forces orientation should be seen as a readout of the internal acto-myosin organization, which confirms that different degrees of polarization were indeed obtained on the different patterns. Strikingly, we observe that centriole-to-centriole axis follows this mechanical polarization (correlation shown in the new Fig.2F). This indicates that the global acto-myosin organization at the cell scale controls the forces at the origin of centrioles separation. Although we could not measure directly centriolar forces, we showed that they are controlled by the global acto-myosin organization of the cell, which we assessed by traction force measurement.

3. The authors conclude that actin contractile forces play a direct role in centrosome separation. To me, this implies that in normal cells, actin contractility is required for centrosomes to separate. I think this is overstating the data. Several studies have indicated that several mechanisms contribute to centrosome separation, and the authors data seems to suggest that in cells without a strong traction axis the effect of perturbing the actin is relatively mild (Figure S5C): it is only in the cells with a high traction axis that actin has a role in driving the greater distancing of the centrosomes that is observed in these cells.

Reply: Although we agree that several mechanisms contribute to centriole separation, we do believe that our data makes a strong case for the role of acto-myosin forces in centriole separation, not only through their magnitude but also their organization. In particular, we showed that:

- In non-patterned cells, inhibiting contractility by blebbistatin reduces the centriole-to-centriole distance (Fig1).
- The centriole-centriole direction correlates with the main acto-myosin force direction on the patterns (Fig2).
- Shapes inducing more acto-myosin force polarisation (H and T) can separate centrioles on larger distances and longer times (Fig3)
- Inhibiting contractility with blebbistatin or ML7 erase the difference between shapes (Fig.4, S6,S7).

We would like to stress that the observed differences between the more polarised shapes (H and T) and less polarised shapes (Square and Tripod) mostly arise from the difference in organisation of acto-myosin fibers rather than their mere contractility, since we did not observe a lower total force magnitude on Square and Tripod. In Square and Tripod, the centrosome is connected to fibers with different directions, so that the forces exerted by some are cancelled by those from others. In H and T, with the same contractility, the acto-myosin fibers are better aligned so that their forces add up constructively to efficiently separate the centrioles. This explains why the blebbistatin treatment has a large effect on centriole separation on H and T, while it has almost no consequence on Square and Tripod (new Fig S7). On those shapes, whatever the contractility, the resulting forces on the centrioles are low from lack of order.

4. The authors conclude that mechanical polarisation regulates centrosome duplication (Figure 4). I think that this is again too strong. I am willing to believe that one can perturb centrosome duplication by forcing cells to adopt fixed geometries, but this is not the same as showing that mechanical polarisation influences centrosome duplication in normal, unperturbed cells.

Reply: In any part of an organism, cells are always confined to a constraint: either solid or liquid tissue. This constraint is dictated by the surrounding cells, the presence of a basal lamina, organ shape, presence of other organs, or shear stress imposed by blood or other fluid flow. In the past decades it has been well documented that geometric constraint can influence cell functions, including cell proliferation (J. Folkman, A. Moscona, Role of cell shape in growth control, Nature). As presented in the Result section (regarding Fig2 and the description of micropatterns), geometrical constraints directly induce mechanical polarization by directing the organization of the acto-myosin cytoskeleton (Mandal et al, Nature Com, 2014). Nevertheless, no direct evidence of mechanical polarization on centrosome duplication has ever been presented. We believe that one piece of evidence might be helping the reviewer understanding our point better: firstly, in the paper by Streichan et al (Spatial constraints control cell proliferation in tissues, Sebastian J. Streichan, Christian R. Hoerner, Tatjana Schneidt, Daniela Holzer and Lars Hufnagel, PNAS) it is well documented that physical barriers – made by PDMS – induce G1/S arrest. G1/S is indeed the checkpoint prior of DNA and also centrosome duplication. This data suggests that geometric constraint in a layer of cell controls the moment of DNA and centrosome regulation. In the same direction, our data show that mechanical polarization induced by geometric constraint impacts cell duplication time (as in the case of Square. We added statistics for this experiments in Fig S10) and efficiency.

Moreover, some of this data is poorly documented. Figure 4A is headed "time from Thymidine release" and two time points are shown "0h" and "Duplication". On reading the legend it seems that this second "Duplication" timepoint is the time at which "all the shapes reach 70% of centrosome duplication". I am not sure exactly what this means or how this was scored (especially as the authors are claiming that many cells have extra centrosomes). Moreover, it seems possible to me that the cells might have different cell cycle dynamics, so is each shape scored independently (in which case this second time point might be a different actual time point for each cell shape) or is the 70% figure an amalgamation of all the cells (in which case the different cell shapes might be at different time points in their cell cycles).

Reply: We thank the reviewer for the correct observation. Different geometric constraints have indeed an impact on the cell cycle time. In particular, Square show a significant delay in centriole duplication: whereas cells on Tripod-T-H do reach the point where 70% of the population have 4 centrioles or more in 10 hours; Square reaches this 70% point only at around 18hours from Thymidine release. We have explained this in the text now and we have performed statistical analyses on the centriole duplication time-course presented in Fig S10, to show that cells on square patterns struggle in duplicating centrioles.

5. Another potential problem with this centrosome duplication analysis is that it is well documented that centrin-GFP can form transient assemblies, that are not centrioles, in cells under certain conditions. Perhaps mechanical polarisation can influence the tendency of centrin-GFP to form these transient dots? I think the authors need to prove more thoroughly that these centrin-dots are really bona-fide centrioles (ideally with EM, but at the very least with other markers, and by showing that these extra centrosomes are stably inherited by the daughter cells after cell division).

Reply: **This point was raised by reviewer1 as well. This is why we attach here the same reply.** We have now added better characterization of these counted centrin1 dots: we have added a completely new figure (FigS13A-C), showing that about 80 per cent of the dots counted are decorated by PLK4. As stated in "Centriolar satellite- and hMsd1/SSX2IP-dependent microtubule anchoring is critical for centriole assembly" by Hori et al (Mol Biol Cell, 2015), centriole satellites do not recruit PLK4, indicating that PLK4 can be used as a centriole marker to identify satellites. From our experiments we observed that some of the counted spots were centriole satellites (20% per cent of the cells), but that this percentage was not different among the shapes analysed, suggesting that the differences in centriole number observed among shapes with high vs low acto-myosin order is not due to the presence of more centriole satellites, but rather to over duplication of centrioles.

Minor Points:

1. I was confused about the data showing the quantification of centrosomal Plk4 levels. In Figure 5A the images all show relatively high levels of centrosomal Plk4, and quantify the strong decrease in Plk4 levels observed after various drug treatments. In Figure S9 (mistakenly referred to as 5E in the text?), there is very little Plk4 detected in any of the images shown, and the authors show that the amount of centrosomal Plk4 actually depends on the position of the centrosome within the cell. Why the difference in these images, and how do the authors account for this positional difference when they quantify the amount of Plk4 at the centrosomes in Figure 5A?

2. The signal to noise seems very low in the Y cells shown in Figure 2A. This should be commented on.

Reply to point1 and2:

We believe we have improved the way to present this point, by explaining in the main text and in the "material and methods" section the strategy used to quantify PLK4 at the centrosome (considered as the mean of the two centrioles area).

We removed the previous Fig. S9 and change the data presentation. We believe that the origin of the confusion lays in the previous Fig. S10: This plot did not show the intensity versus the distance between centrioles or their position, but it aimed to show the level of PLK4 at difference distances from the centriole signal centre, to indicate that the PLK4 value we measured is really at the centriole area.

We believe that PLK4 recruitment is not driven by centriole separation, but rather that both processes are controlled by acto-myosin force organization. Fig 6 shows that more ordered actin organization induces lower level of PLK4 recruitment

Since we are here talking of both centrioles, we referred to them as the centrosome. The value measured is the mean intensity, which is the intensity normalized by the area selected. This probably explains the "low" values seen by the reviewer. We did not present in the chart the noise level, which perhaps it was the reason of the confusion.

We believe that this version would more clearly convey the message: what we show here is that the acto-myosin organization, not the position of the centriole, influences the recruitment of PLK4 at the centrosome. We added a model in the end of the paper, with the idea that it could help the reader understand the logic of the paper and the results (Fig 8).

In summary, even if the authors can address these issues, I would advise a more caution in their headline claims.

Reply: we have changed headlines of figures and paragraph to avoid overstatements.

These findings might suggest that cells in which mechanical polarisation has been imposed in this relatively artificial system have a tendency to separate to a greater degree and perhaps to over-duplicate, but this does not mean mechanical polarisation normally plays a major part in regulating centriole separation and duplication in normal tissues.

Reply: To answer the reviewer's concern about the artificial nature of our system, we have added a Figure1 to present how the situation really is not different on "normal" non-patterned cells: we observe that HeLa cells can separate centrioles with the same distance range as shown by the cells on adhesive patterns. As the opening of our paper, these data introduce the interest of using of micropattern to modulate acto-myosin force polarization and investigate its relationship with centriole separation. Hence we believe what we observed are not only induced by the artificial system imposed to the cells.

Thus, the results reported here are an interesting first step in understanding the potential link between mechanical polarisation and centrosome behaviour, but they do not prove this link normally plays a major role in regulating centrosome behaviour in normal cells.

Reviewer #3 (Remarks to the Author):

This paper is very poorly written and difficult to follow because of its careless use of language in the main body of the text. Throughout the paper, we are told that "cells" respond to various treatments and the reader is only told of the identity of these cells a single time in the Results section.

Reply: We apologize for the confusion caused by our way of describing the experiments and explaining concepts. We now put efforts into making the text easier to read and smooth, expanding paragraph containing explanation of main experiments, for better understanding of the logic behind this paper. We believe that the new shape of this text would help the reader follow the research presented.

We also have been careful in clarifying the cell line used in each section and figure legend.

This is not a minor point, as the study focuses on one tumour cell type and so its generality is uncertain. Moreover, phenomenon are described that we told are affecting centrosomes or centrioles interchangeably ...which? And the phenomenon affected is either centrosome / centriole duplication or sometimes amplification. I am concerned that this lack of precision extends into the interpretation of the data which is largely correlative in its nature.

Reply: We chose to use HeLa cells for this experiment, since it is a common model widely used in cell division studies. Moreover, the great availability of stable cell lines produced in HeLa cells (in particular for centriole components) made the HeLa cells an attractive model to utilize to prove our hypothesis. We believe that it would be interesting to test the observed phenomena with other cell lines, preferably not-transformed.

We apologize for the confusion about centriole duplication/amplification. The other reviewers did not seem having the same issue in the usage of these terms but we think we changed the text in order to explain better what we mean for aberrant centriole duplication by stating in each related paragraph that we considered over-amplified a number of centrioles corresponding to more than 4C. Alongside centriole amplification, we counted the frequency of centriole duplication too, which we defined as 4 centrioles or more. This count quantifies the duplication rate and is different from the over-amplification rate (more than 4C) that represent a sign of how good the cells have duplicated the centrioles.

The conclusion of the first section of the paper is that "cells with high mechanical polarization can separate centrioles for larger distances and longer times". While this is certainly one correlation, there are many others that can be made here because the organisation of the HeLa cells is greatly disrupted by their culture on these micropatterns.

Reply: We have added Figure1 to present how the situation really is not different on micropatterns: we observe that HeLa cells can separate centrioles with the same distance range shown by the cells on adhesive patterns. This is now the opening of our paper and it gives further support to the use of micropattern to investigate better the relationship between acto-myosin forces and centriole separation. This also suggests that the results observed are not only induced by the artificial system imposed to the cells.

This question of interpretation extends to the second section that uses Blebbistatin to block myosin II. It reaches a conclusion that actin contractility is required for centriole separation. Here perhaps the authors mean centrosome separation as centriole separation is the disengagement process that occurs in early G1. Notwithstanding the question of what they mean, the observation is again only correlative and does not indicate whether the effect observed is direct or indirect. This same question of interpretation arises with the analysis of the relationship between inter-centriolar (do the authors mean centrosomal?) distance and cellular traction.

Reply: We apologize for the confusion regarding the centrosomal distances. The experiments presented in this paper are all dedicated to the characterization of centriole-to-centriole distances. We have now favoured in the paper the expression "centriole-to-centriole distance" rather than intercentriolar distances, to prevent confusion. We also commented on the finding showing that specific distances seem to be specific to precise moments of the cell cycle: while this paper was in revision, a new paper came out reporting that cell contractility increases during G1 till S phase. Here it reaches a plateau, for then decreasing during G2 phase (Variation in traction forces during cell cycle progression. Vianay B, Senger F, Alamos S, Anjur-Dietrich M, Bearce E, Cheeseman B, Lee L, Théry M., Biol Cell. 2018). These data now reinforce our work by showing that contractile forces vary through cell cycle, making easier for us to advance the hypothesis that different centriole-to-centriole distances are specific to particular phases of the cell cycle.

The authors also attempt to test the relationship between actin generated forces and "centriole duplication number".... Although I don't understand what they mean here. Here they show, not unexpectedly, that centriole/centrosome separation depends upon the cell cycle phase placing some degree of doubt upon their earlier experiments with unsynchronised cells. Their conclusions about the fidelity of centriole duplication go beyond the reasonable interpretation of their data as they do not examine the centriole duplication cycle in any way whatsoever.

Reply: In the first part of the paper, the observation on the asynchronous cells are indeed a first observation of the impact of acto-myosin forces on centriole separation. Cells coming from the same population were plated on the different shapes. Since all other conditions are identical, it is the geometric constraint that triggers the differences we observed. In the first half of the paper, our experiments on asynchronous cells show that organization of acto-myosin forces affects centriole separation and we observed a correlation between centriole separation and mechanical force. In the second part of the paper, we indeed wondered if cell cycle phases contribute to the variability of centriole-to-centriole distances. This is why we then worked with cells synchronized in G1: this way, we made sure that all the cells were in the same starting point, at the moment of the experiments. In this experiment, the results observed show again a significant correlation between centriole separation and mechanical polarization, although the particular set of distances (2-6µm) is not observed now, suggesting it belongs to perhaps very early stages of G1.

They then attempt to draw conclusions based upon the enrichment of Plk4 at the centrosomes (Do they mean centrioles?)

Reply: We believe we have improved the way to present this point, by explaining in the material and methods the strategy used to quantify PLK4 at the centrosome (considered as the sum of the two centrioles area, hence the total PLK4 recruited at both centrioles. Since we are talking about both centrioles, we referred to them as the centrosome.

without any regard of the existing knowledge of Plk4 recruitment in the centriole duplication cycle. The authors should relate their findings to the known ability of Plk4 to re-localize from a ring around the outer part of the centriole to a single dot like structure on procentriole formation – the key step in the centriole duplication cycle and whether the effects they observe are similar, for example, to those seen when Plk4 levels increase as a consequence of inhibition of the SCF complex.

The final section about potential roles of Arp2/3 in Plk4 recruitment is simply unconvincing.

Reply: **This point was raised by reviewer1 and 2 as well. This is why we attach here the same reply.** We present here a version of the paper that focused on PLK4 with no reference to Arp2/3. Data on Arp2/3 and its involvement in PLK4 mediated centriole duplication in response of acto-myosin forces is still preliminary and it could be the subject of a dedicated project. However, we mention in the discussion that there are evidences in the literature suggesting that it would be

interesting to focus on Arp2/3 role in PLK4 mediated centriole duplication and that this could be a perspective of continuation of this project.

In conclusion, the authors should re-evaluate their manuscript both with respect of the precision with which they refer to the structures being studied and the level of analysis of functions for which there is already a very solid basal level of knowledge to which any new findings should be referred.

Reply: We agree with the reviewer on this point. We have amply changed the text, clarified the message, added appropriate referenced, documented more unclear points, and removed confusing statements. We now believe this new version will meet the reviewer's expectations.

Yours truly,

Elisa Vitiello, PhD

Martial Balland, PhD

Reviewers' comments:

Reviewer #1 (Remarks to the Author):

In their revised article, Vitiello and colleagues have answered most of my concerns. In particular, they have simplified their article by focusing on the link between mechanical polarization of cells and centrioles separation and duplication and removing the last part, which was less clear and poorly supported by the data. They also included a few interesting new experiments and analysis, such as the comparison with non-patterned cells. Finally, they clarified the writing and the figures. There are still a few minor points, which are not very clear, and the need for eliminating typos from the manuscript, but overall I find the manuscript suitable for publication.

1) In figure 5, with the thymidine bloc release, it is really not very clear to me which centrin dots the authors count as centrioles. They might add, at least in supplementary data, a larger version of the image of the centriole with a semi-transparent colored dot placed on the centrin dots considered as centrioles. It is almost certain that some of the extra dots are not centrioles and it is not clear how the authors distinguish between them.

2) In figure 6: there is something a bit complicated with the cenB treatment: if the authors quantify the decrease in PLK4 recruitment in, for example, T versus Tripod. In control cells, they would find a value below 1 (decrease on the Tripod), but if they compare the staining of cells treated with CenB, the ratio should rather be one or slightly larger. This would mean that CenB has an effect, as it kills the difference between T and Tripods. But as they present the data, the difference, for each shape, between control and CenB treated cells, is not significant, meaning there is no effect of CenB..Could they do something about that and clarify whether or not there is a significant effect of CenB on PLK4 recruitment?

Reviewer #2 (Remarks to the Author):

In this revised manuscript the authors have gone some way to addressing my main concerns, but I still think two issues remain unresolved and so I cannot support publication.

First, I was concerned that the authors overstate the importance acto-myosin forces in controlling centriole duplication. The authors show very nicely that these forces can influence the timing of centriole separation, and the distances that the separating centrioles move apart—but this is very different from concluding that the cell normally regulates these forces to control centriole duplication. I would have been happy if this conclusion had just been toned down, but the authors have chosen not to do this (this is still a major conclusion in the title and abstract). I'm afraid I do not find their arguments to counter this point (in the paper or their response letter) very convincing. Blebbistatin strongly blocks centriole separation, but the authors do not prove that it has blocked duplication.

Second, I asked the authors to prove that the extra centrin dots they observe in Square and Tripod cells are really extra centrioles. They claim to have now done this but, if this is the case, I was very confused by the presentation of this data. In Figure 5 the authors show extra centrin dots in Square and Tripod cells compared to H and T cells (quantified in 5B). They then argue that these dots are really extra centrioles (rather than centriolar satellites) because they contain Plk4. This data is shown in Figure S13. But this Figure does not seem to compare centrin and Plk4 dots in cells arrested in G1 with a double thymidine block (as shown in Figure 5), but rather seems to compare these in untreated cells. Moreover, I think the authors are arguing that the untreated Square and Tripod cells still show extra centrin dots (quantified in S13B, but this data is not statistically analysed), but do not show extra Plk4 dots. Perhaps this is not what they are trying to show here but, regardless, the key experiment is to show that the extra centrin dots they observe in the G1 arrested cells (as shown in Figure 5B) all contain Plk4. I could not see this data in the paper and so I remain sceptical that the authors are really observing bona fide centriole

overduplication here. Thus, at present, I think one of the major conclusions of the paper is not supported by the data.

Reviewer #2's comments on Reviewer #3's previous report:

It looks to me like this reviewer shared several of my concerns. On the major point that the authors claim their work sheds light on the mechanisms of centriole duplication (rather than just centrosome separation) I doubt this reviewer would be convinced by the authors changes (I certainly wasn't). I don't think the authors have done any new experiments to address the final point about Arp2/3, which I agree was weak. In my opinion, the authors have not done enough to address these concerns.

Reviewer #1 (Remarks to the Author):

In their revised article, Vitiello and colleagues have answered most of my concerns. In particular, they have simplified their article by focusing on the link between mechanical polarization of cells and centrioles separation and duplication and removing the last part, which was less clear and poorly supported by the data. They also included a few interesting new experiments and analysis, such as the comparison with non-patterned cells. Finally, they clarified the writing and the figures. There are still a few minor points, which are not very clear, and the need for eliminating typos from the manuscript, but overall I find the manuscript suitable for publication.

1) In figure 5, with the thymidine bloc release, it is really not very clear to me which centrin dots the authors count as centrioles. They might add, at least in supplementary data, a larger version of the image of the centriole with a semi-transparent colored dot placed on the centrin dots considered as centrioles. It is almost certain that some of the extra dots are not centrioles and it is not clear how the authors distinguish between them.

Both reviewers raised this concern (Reviewer #2's second question) about whether the extra-dots observed in low mechanically polarized cells (Square and Tripod) are extra-centrioles or satellites.

To discriminate extra-centrioles from centriolar satellites, we used PLK4 staining in a systematic way, after thymidine block release. However, as shown by others in the literature (Hori, A., Peddie, C. J., Collinson, L. M. & Toda, T. Centriolar satellite- and hMsd1/SSX2IP-dependent microtubule anchoring is critical for centriole assembly. *Mol. Biol. Cell* 26, 2005–2019 (2015)), PLK4 labelling does not lead to well-resolved dots (but rather clouds) and do not necessarily colocalize with the C1-labeled centrioles. Therefore, it did not allow to determine unequivocally which dots are bona fide centrioles: dots are regularly found at the frontier of the PLK4 cloud, leading to uncertainty as to whether they should be counted as centrioles or satellites. Hence, we could not experimentally prove that each counted dot is an extra-centriole. We decided to change the manuscript perspective by presenting the C1-dots counts in the perspective of PLK4 inhibition. PLK4 is a widely-recognized player in centriole over-amplification and its inhibition with CenB has been validated numerous times and is even used in cancer therapy clinical trials (Mason, J. M. et al. Functional characterization of CFI-400945, a Polo-like kinase 4 inhibitor, as a potential anticancer agent. *Cancer Cell*, 2014) (Ines Lohse, Activity of the novel polo-like kinase 4 inhibitor CFI-400945 in pancreatic cancer patient-derived xenografts, *Oncotarget*, 2017). Since PLK4 inhibition has been shown to cause reduction of authentic centrioles and only dispersion but not elimination of centriole satellites (Akiko Hori et al, *A non-canonical function of Plk4 in centriolar satellite integrity and ciliogenesis through PCM1 phosphorylation* EMBO reports 2016), our observation that the number of cells presenting more than 4 C1 dots in Square and Tripod is strongly reduced by PLK4 inhibition proves that the extra centrioles observed in our experiments are not centriole satellites and that the mechanism at play in those shapes seems indeed to be centriole amplification.

The finding that PLK4 inhibition can rescue the lack of mechanical polarization is, in the context of cancer biology, a result that deserves further investigation. Showing that PLK4 inhibition can rescue the lack of mechanical polarization is quite a strong result and that it is more relevant than the question of whether some of the dots are centriolar satellites, which existence and stainings as different entities from centrioles is still under debate (Hori, A., Peddie, C. J., Collinson, L. M. & Toda, T. Centriolar satellite- and hMsd1/SSX2IP-dependent microtubule anchoring is critical for centriole assembly. *Mol. Biol. Cell* 26, 2005–2019 (2015). While we admit that some of the C1 dots may be satellites (and refer to Fig S13 which suggests that 20% to 30% of observed centrioles may be satellites, over a population of cells), the fact that PLK4 inhibition, known to prevent centrioles over-amplification, drastically reduces the number of dots in cells with low acto-myosin polarization, strongly suggests that the number of C1 dots is a valid indicator of centriole over-amplification.

2) In figure 6: there is something a bit complicated with the cenB treatment: if the authors quantify the decrease in PLK4 recruitment in, for example, T versus Tripod. In control cells, they would find a value below 1 (decrease on the Tripod), but if they compare the staining of cells treated with CenB, the ratio should rather be one or slightly larger. This would mean that CenB has an effect, as it kills the difference between T and Tripods. But as they present the data, the difference, for each shape, between control and CenB treated cells, is not significant, meaning there is no effect of CenB...Could they do something about that and clarify whether or not there is a significant effect of CenB on PLK4 recruitment?

We agree that the message in this figure was confusing: our aim was to focus on the recruitment of PLK4 which are shown to be different between shapes. Centrinone B treatment has been applied to make sure that inhibiting the kinase activity of PLK4 would not affect its ability to be recruited at the centrosome, since this drug would be used in the next step (former Fig.7) where we show that, by inhibiting PLK4 activity, the number of centrosome misduplication can be reduced.

As the reviewer pointed out, it does seem that the difference of PLK4 recruitment between the shapes are reduced after Centrinone B treatment. However, we cannot conclude, with statistical significance, that CenB treatment causes the disappearance of this difference (as it is a second order effect). We believe more work would be required to reach a reliable conclusion, which is out of the scope of the present manuscript. Here, we use CenB to inhibit PLK4 activity (which is a demonstrated effect in literature (Mason, J. M. et al. Functional characterization of CFI-400945, a Polo-like kinase 4 inhibitor, as a potential anticancer agent. *Cancer Cell*, 2014)). If it has a differential impact on PLK4 recruitment for different shapes, we believe it to be much weaker and hence negligible at first order. To avoid confusion in the manuscript, we removed the quantification of PLK4 fluorescence intensity with CenB treatment in Fig. 6, since it is not the main message. Instead, we mention in the text that CenB did not perturb significantly PLK4 recruitment and refer to data in supplementary (Fig. S15).

Reviewer #2 (Remarks to the Author):

In this revised manuscript the authors have gone some way to addressing my main concerns, but I still think two issues remain unresolved and so I cannot support publication.

First, I was concerned that the authors overstate the importance actomyosin forces in controlling centriole duplication. The authors show very nicely that these forces can influence the timing of centriole separation, and the distances that the separating centrioles move apart—but this is very different from concluding that the cell normally regulates these forces to control centriole duplication. I would have been happy if this conclusion had just been toned down, but the authors have chosen not to do this (this is still a major conclusion in the title and abstract). I'm afraid I do not find their arguments to counter this point (in the paper or their response letter) very convincing. Blebbistatin strongly blocks centriole separation, but the authors do not prove that it has blocked duplication.

We thank the reviewer for the relevant comments. We now have extensively taken into account this reviewer's recommendation to narrow our claims, to the satisfaction of the first reviewer. In the revised version presented here, we toned down again our claims, as explained above, by being more careful with the notion of centriole misduplication (which was removed from the title and headlines). However, we feel that presenting the consequence of mechanical polarization beyond centriole separation is essential in this story, as it brings some insight into the **functional**

relevance of the centriole separation process. Centriole separation in itself would not give a complete picture of the role of acto-myosin forces in centrosome regulation, which we believe to hold a huge potential interest for aneuploidy and cancer prevention. As added in the main “Results” part and in the “Discussion”, our results are in agreement with what was previously shown by Shukla and others, where centriole separation might work as a mechanism to prevent centriole duplication (Shukla, A., Kong, D., Sharma, M., Magidson, V. & Loncarek, J. Plk1 relieves centriole block to reduplication by promoting daughter centriole maturation. Nat. Commun. 2015).

In the revised manuscript, we start by studying the effect of acto-myosin force organization on cell cycle progression, after which we focus on centriole duplication. For that reason, we studied the cell cycle more globally. We introduced new experimental results on how acto-myosin force polarization affect cell cycle timing, using FUCCI cells. The duration of the S and G2 phases were increased for cells under low mechanical polarization or submitted to blebbistatin treatment. Since this stage of the cell cycle corresponds to the time when centrioles duplicate and this is also associated with PLK4 activity, we then focused on the recruitment of PLK4 to the centrioles. Interestingly, we found differences in PLK4 recruitment to the centrioles for the different shapes. These results have been added to the manuscript. In agreement with the reviewer’s concern, we no longer present the quantification of C1 dots as a direct proof of centriole over-amplification but rather with respect to PLK4 inhibition (which was found to rescue the effect of low acto-myosin order). PLK4 inhibition has been shown to cause only centriole satellite dispersion but not elimination. Hence, the reduction of extra C1-dots in shapes with low mechanical polarization indicates that these dots are authentic centrioles (Akiko Hori et al, *A non-canonical function of Plk4 in centriolar satellite integrity and ciliogenesis through PCM1 phosphorylation* EMBO reports 2016). These data taken together form a very consistent body of evidence that all point to a role of acto-myosin polarization in the centriole duplication process.

We rewrote the text of the manuscript (paragraph “Acto-myosin forces regulate S-G2 phase, PLK4 recruitment and activity to limit centriole duplication”) to present this data in a straightforward way and avoid claims that may that may not be fully supported by the evidences at hand. The data with FUCCI cells are presented in Fig5 and S10. Experiments on PLK4 recruitment and inhibition were rearranged in new Fig6.

Second, I asked the authors to prove that the extra centrin dots they observe in Square and Tripod cells are really extra centrioles. They claim to have now done this but, if this is the case, I was very confused by the presentation of this data. In Figure 5 the authors show extra centrin dots in Square and Tripod cells compared to H and T cells (quantified in 5B). They then argue that these dots are really extra centrioles (rather than centriolar satellites) because they contain Plk4. This data is shown in Figure S13. But this Figure does not seem to compare centrin and Plk4 dots in cells arrested in G1 with a double thymidine block (as shown in Figure 5), but rather seems to compare these in untreated cells. Moreover, I think the authors are arguing that the untreated Square and Tripod cells still show extra centrin dots (quantified in S13B, but this data is not statistically analysed), but do not show extra Plk4 dots. Perhaps this is not what they are trying to show here but,

regardless, the key experiment is to show that the extra centrin dots they observe in the G1 arrested cells (as shown in Figure 5B) all contain Plk4. I could not see this data in the paper and so I remain sceptical that the authors are really observing bona fide centriole overduplication here. Thus, at present, I think one of the major conclusions of the paper is not supported by the data.

As presented above in response to reviewer #1's similar concern, we agree with the reviewer that there may be 'satellites' in the C1 dots observed. Therefore, in the newly revised manuscript, we explicitly take into account this possibility. However, this does not fundamentally call into question the conclusions of this work, since many of our experimental data point in the same direction (see answer to reviewer #1 question 1).

Reviewer #2's comments on Reviewer #3's previous report:

It looks to me like this reviewer shared several of my concerns. On the major point that the authors claim their work sheds light on the mechanisms of centriole duplication (rather than just centrosome separation) I doubt this reviewer would be convinced by the authors changes (I certainly wasn't). I don't think the authors have done any new experiments to address the final point about Arp2/3, which I agree was weak. In my opinion, the authors have not done enough to address these concerns.

We agree with the reviewers concerns regarding the role of Arp2/3 in centriole duplication. For that reason, we decided to highlight the effect of acto-myosin force organization on centrosome separation and remove the more preliminary Arp2/3 data. With this, we hope to present a more consistent global picture of its consequences (in terms of cell cycle, PLK4 recruitment and centriole amplification), that could convince the reviewers of the conclusions of our manuscript.

To clarify the message, we also revised the data presented in supplementary. We removed the following figures that are irrelevant for the new version and potentially confusing:

-speed of centriole separation

We merged supplementary data and main figure data (former S12 in new S6E,F and former S15 in new 6H):

- Inhibition of acto-myosin contractility reduces centriole duplication rate (4C and more than 4centrioles all together) (new 6 E, F)
- Inhibition of PLK4 activity reduces centriole duplication rate (4C and more than 4centrioles all together) (new 6 H)

REVIEWERS' COMMENTS:

Reviewer #1 (Remarks to the Author):

In this second revision of their article, the authors have mostly revised their text to address the comments and avoid over-interpreting their results. I think the paper is now suitable for publication, even if they did not solve the point about centriole satellites, because this is now more clearly explained in the text and not overinterpreted.

Reviewer #4 (Remarks to the Author):

I think the authors could go a bit further to solidify this conclusion.

In other words, they should show bona fide defects in duplication by co-staining of markers and perhaps even some EM experiments. The latter is most definitive.

I agree with the reviewers that in this respect, the study fell short.

Without a clear demonstration of an impact on centriole duplication, as described above, it would be difficult to make a case that it is of broad interest

REPLY to REVIEWERS' COMMENTS

Reviewer #1 (Remarks to the Author):

In this second revision of their article, the authors have mostly revised their text to address the comments and avoid over-interpreting their results. I think the paper is now suitable for publication, even if they did not solve the point about centriole satellites, because this is now more clearly explained in the text and not overinterpreted.

We thank Reviewer1 for its input and suggestion and we are happy to have met his expectations.

Reviewer #4 (Remarks to the Author):

I think the authors could go a bit further to solidify this conclusion.

In other words, they should show bona fide defects in duplication by co-staining of markers and perhaps even some EM experiments. The latter is most definitive. I agree with the reviewers that in this respect, the study fell short.

Without a clear demonstration of an impact on centriole duplication, as described above, it would be difficult to make a case that it is of broad interest

We agree with Reviewer4 about the necessity of adding extra centriolar staining to further prove the bona-fide centriole duplication. Electron Microscopy experiments would be interesting. The limiting point in this set-up is that cells are plated on micropatterns that confine the cells to particular geometries, in order to induce particular actin organization responses. EM has fixation and sample slicing protocols that risk harming the micropattern and the shape of the cell itself, with the possible consequent alteration in centriolar detection. This represents in fact the limit of the technique. Yet, improvement of the technique would be required in the future.

We have now added in the Discussion a section presenting the importance of more extra centriolar staining and high-resolution techniques such as EM imaging. Moreover, as required by the Editor, we toned down the conclusions of the experiments in Figure 6 regarding the impact of actin organization on centriole duplication.